# Bridging the gap: a new module for human water use in the Community Earth System Model version 2.2.1

Sabin I. Taranu[1], David M. Lawrence[2], Yoshihide Wada[3,4], Ting Tang[3,4], Erik Kluzek[2], Sam Rabin[2], Yi Yao[1], Steven J. De Hertog[1,5], Inne Vanderkelen[6,7,8], and Wim Thiery[1]

[1]Vrije Universiteit Brussel, Department of Water and Climate, Brussels, Belgium
[2]National Center for Atmospheric Research, Climate and Global Dynamics Laboratory, Boulder, CO, USA
[3]Biological and Environmental Science and Engineering Division, King Abdullah University of Science and Technology, Thuwal, Saudi Arabia
[4]International Institute for Applied Systems Analysis, Laxenburg, Austria
[5]Universiteit Gent, Department of Environment, Q-ForestLab, Ghent, Belgium
[6]Wyss Academy for Nature at the University of Bern, Bern, Switzerland
[7]Climate and Environmental Physics division, University of Bern, Bern, Switzerland
[8]Oeschger Centre for Climate Change Research, University of Bern, Bern, Switzerland

**Correspondence:** Sabin I. Taranu (sabin.taranu@vub.be)

**Abstract.** Water scarcity is exacerbated by rising water use and climate change, yet state-of-the-art Earth system models typically do not represent human water demand. Here we present an enhancement to the Community Earth System Model (CESM) and its land (CLM) and river (MOSART) components by introducing sectoral water abstractions. The new module enables a better understanding of water demand and supply dynamics across various sectors, including domestic, livestock, thermoelectric, manufacturing, mining, and irrigation. The module conserves water by integrating abstractions from the land component with river component flows, and dynamically calculates daily water scarcity based on local demand and supply. Through land-only simulations spanning 1971–2010, we verify our model against known water scarcity hotspots, historical global water withdrawal trends, and regional variations in water use. Our findings show that non-irrigative sectoral consumption has an insignificant effect on regional climate, while emphasizing the importance of including all sectors for water scarcity assessment capabilities. Despite its advancements, the model's limitations, such as its exclusive focus on river water abstractions, highlight areas for potential future refinement. This research paves the way for a more holistic representation of human-water interactions in ESMs, aiming to inform sustainable water management decisions in an evolving global landscape.

## 1 Introduction

Human-induced land-use modifications together with water resources management have substantially impacted the Earth's surface and modified the terrestrial water cycle (Foley et al., 2005; Oki and Kanae, 2006; Wada et al., 2010; Rodell et al., 2018). Water serves multi-functional purposes for humans, including agriculture, industrial processes, domestic use, and ecological services. Over the past century, there has been a significant shift in total human water use, driven primarily by factors such as population growth, industrialization, and urbanization (Shiklomanov, 2000; Vörösmarty et al., 2000; UNESCO, 2021).

Towards the future, these drivers are projected to further evolve, influenced by technological advancements, socio-economic
transitions, and changing water use patterns, thereby likely leading to heightened water demand (Wada et al., 2016). Already,
regions globally grapple with issues of drought and water insecurity, challenges that underscore the criticality of sustainable
water management (Hoekstra et al., 2012; Wada et al., 2013; Trenberth et al., 2014; Famiglietti, 2014; Mekonnen and Hoek-
stra, 2016; Kummu et al., 2016). Adding complexity to this scenario, anticipated climatic changes, characterized by variable
precipitation patterns and altered hydrodynamics, will further strain water supply systems (Milly et al., 2008; Trenberth, 2011;
Döll and Schmied, 2012; Arnell and Lloyd-Hughes, 2014).

Deforestation and urbanization not only perturb carbon dynamics but also profoundly alter the hydrological cycle, compro-
mising water availability and quality (Coe et al., 2009; Pan et al., 2011; Seto et al., 2012; Baccini et al., 2017). The expansion
of agriculture, driven by human food demands, modifies natural catchments and exacerbates water withdrawals, placing im-
mense stress on both surface and groundwater resources (Wada et al., 2012; Famiglietti, 2014; de Graaf et al., 2019). Such
over-extraction has led to phenomena such as land subsidence and saltwater intrusion, directly threatening the sustainability
of freshwater sources (Bierkens and Wada, 2019). The construction of dams and reservoirs, while providing water storage
and energy benefits, disrupts riverine ecology, impacts sediment and nutrient transport, modifies natural flow regimes, and
can impact local climates (Grill et al., 2015; Best, 2019; Vanderkelen et al., 2021). Pollution, another by-product of human
activity, notably from untreated wastewater, poses dire health risks and compromises the integrity of freshwater ecosystems
(Vörösmarty et al., 2010). Concurrently, wetland drainage and land reclamation, often undertaken to meet human settlement
or agricultural demands, diminish the natural buffering and filtration capacities inherent to these systems (Davidson, 2014).
Collectively, these human-driven changes to the land and water nexus not only perturb biogeochemical cycles but also have
significant implications for water security, human health, and the socio-economic stability of communities (Bakker, 2012; Link
et al., 2016).

The study of the water cycle and broader Earth system changes relies critically on advanced modelling frameworks. Among
them, the most integrated are the Land Surface Models (LSMs), which represent the land surface within Earth System Mod-
els (ESMs) (Blyth et al., 2021). LSMs primarily simulate the interaction between the terrestrial biosphere, atmosphere, and
hydrosphere, capturing processes like evapotranspiration, soil moisture dynamics, and snow accumulation and melt. Comple-
menting this, ESMs encapsulate a more comprehensive set of processes and interactions, including atmospheric, oceanic, and
cryospheric components, which allow for a holistic examination of the Earth's climatic and environmental dynamics. These
models operate over various temporal and spatial scales. At the finer end, some models have spatial resolutions as detailed as
a few kilometres, making short-term weather predictions and analysing specific hydrological processes possible (Prein et al.,
2015). Conversely, coarser resolutions spanning up to hundreds of kilometres are suitable for long-term climate projections
over centuries or millennia (Eyring et al., 2016). Leveraging these scales, researchers can explore a spectrum of phenomena,
from short-term flood events to long-term climate change impacts, and from localized water table shifts to global sea-level
rise (Drijfhout et al., 2015; Jevrejeva et al., 2016; Liu et al., 2018; Mankin et al., 2019; Wu et al., 2020). The adaptability and
robustness of these frameworks provide invaluable tools for scientists aiming to understand and predict changes in the Earth's
water cycle and broader environmental systems.

The field of Earth system modelling has seen important progress between different iterations of the Coupled Model Intercomparison Project phases, demonstrating higher skill in matching observations across relevant climate change indicators (IPCC AR6 Technical Summary: The Physical Science Basis, 2021, see Fig. TS.2). This can be attributed to running the models at higher resolutions, increasing model complexity, and improving the representation of physical, chemical, and biological processes (IPCC AR6 Technical Summary: The Physical Science Basis, 2021). Despite the progress made, there are still processes not well captured in ESMs. For example, describing human-water interactions was previously recognized as one of the important challenges in Earth system modelling (Nazemi and Wheater, 2015). Recent years, however, have witnessed targeted efforts to bridge this deficiency. This includes implementation of the land-use and land-cover change (LULCC) (Lawrence et al., 2016), urban surfaces and associated hydrological disturbances (Lipson et al., 2023), irrigation water use (Blyth et al., 2021; McDermid et al., 2023), large dams and flow regulation (Yassin et al., 2019; Vanderkelen et al., 2021, 2022), and groundwater use (Pokhrel et al., 2016; Nie et al., 2018; Felfelani et al., 2021). A more complete representation of human-water interactions, including abstractions for all sectors from both surface and groundwater sources, as well as reservoir operations, to our knowledge is currently operationally available for only one ESM system (i.e. MIROC6; Yokohata et al., 2020).

To further contribute to the effort of improving human-water interactions in LSMs/ESMs, we here present a new sectoral water use module for the Community Earth System Model version 2 (CESM2). Our data-driven module advances the representation of human water use by incorporating a comprehensive account of water abstractions for domestic, livestock, thermoelectric, manufacturing, and mining sectors, thereby complementing the existing irrigation module (Lawrence et al., 2019). Through this development, the CESM2 model and its land component more closely approach the capabilities of state-of-the-art Global Hydrological Models (GHMs), which not only represent essential hydrological processes but also commonly integrate human-related water management practices, including reservoir operations, water abstractions, pollution, and the exploration of alternative water sources like desalination and wastewater reuse (Hanasaki et al., 2016; Sutanudjaja et al., 2018; Hanasaki et al., 2018; Burek et al., 2020; Droppers et al., 2020; Müller Schmied et al., 2021; Van Vliet et al., 2021; Jones et al., 2023). Additionally, it enables fully coupled applications, allowing for the exploration of feedbacks between human water use and land-atmosphere interactions (Keune et al., 2018), which is not achievable with GHMs.

The next section first describes the CESM2 modelling framework with its existing capabilities in representing human-water interactions (Sect. 2.1). Subsequently, a detailed description of the functioning of the new sectoral water use module is given (Sect. 2.2). Next, the information about the prescribed non-irrigative input data is provided (Sect. 2.3). A hypothetical case study is then proposed to better understand the sectoral competition algorithm under limited water availability situations (Sect. 2.4). Since the newly added sectors are data driven and based on previously-evaluated inputs (Huang et al., 2018), the validation section is focused on the robustness of the implementation itself (Sect. 3.1). To assess the capability of the newly developed module, the global and regional trends in sectoral water withdrawal are analysed for the historical period 1973–2010, with distinction being made between expected and actual fluxes (Sect. 3.2-3.3). The ability of irrigation and other sectors' consumption to impact local climates through changes in surface water-energy exchanges is then investigated (Sect. 3.4). Next, model results are used for a global qualitative water scarcity assessment, showing the model's ability to predict

known hotspots of water scarcity (Sect. 3.5). Lastly, before concluding (Sect. 5), a discussion of existing limitations and possible future refinements is provided (Sect. 4).

## 2  Methods

### 2.1  Existing human-water representation in CESM2

The CESM2 is an open-source, community-developed Earth system model that encompasses ocean, atmosphere, land, sea-ice, land-ice, river, and wave models. These individual models, which may operate at different spatial resolution and time step, interact and exchange states and fluxes via a coupler (Danabasoglu et al., 2020). Since human water abstractions occur over land, we have focused our model development on the land and river components of the CESM2 model, which are the Community Land Model version 5 (CLM5; Lawrence et al. (2019)) and the Model for Scale Adaptive River Transport (MOSART; Li et al. (2013)).

CLM5 already models irrigation, with irrigation water abstractions being computed on a daily basis based on soil moisture deficits and irrigated crop water requirements (Sacks et al., 2009; Lawrence et al., 2019). The default source for water supply for irrigation is the river network, with a user-defined possibility to supply from groundwater. At the moment, however, the simulated groundwater abstractions for irrigation are not constrained by observations and this new CLM5 module has not been thoroughly tested. Therefore, in this study, we exclusively use the default configuration where water is abstracted from the river network.

River water availability within CLM5 is provided by the MOSART routing model. It utilizes a kinematic wave approach, providing information on varying channel velocities, water depth in channels, and channel surface water variations (Li et al., 2013). In its functionality, surface runoff from CLM5 first traverses hillslopes before merging with subsurface runoff and moving to a tributary network, finally ending up in the main channel (Fig. 1 from Li et al. (2013)). Each MOSART grid cell has a single main channel that connects the local spatial unit with upstream/downstream units through the river network. It is this main channel's water storage, aggregated at CLM5 grid-cell level, that are used to estimate current river water availability. It should be noted that the CLM5 and MOSART models can run using different grids, which is the case in this study, with MOSART running on a 0.5x0.5° grid, and CLM5 on 0.9x1.25° grid. This means that for a given CLM5 grid cell, several MOSART main channels will be sourced for water supply. The handling of these spatial discrepancies is done through remapping procedures in the coupler.

Once the irrigation water demand is met by abstraction from the river network, it is applied over the surface soil across irrigated crop columns (Fig. A1). This arrangement allows the water to contribute to crop growth and become a part of the surface water-energy balance through processes like evapotranspiration, runoff, and infiltration (Fig. A2). In case there is not enough water to fully satisfy irrigation requirements, the model has 2 options. First, to limit irrigation abstraction to 90% of current river storage, which helps maintain at least 10% for environmental flow requirements. Or secondly, abstract as much as needed, with the shortfall being compensated by ocean water. While less realistic, the second setup was successfully used in studies where having total irrigation requirements satisfied is important (Thiery et al., 2017; Yao et al., 2022).

Recently, additional developments have been completed to advance human water representation in CESM2, notably the dynamically changing open water surfaces to represent historical reservoir construction (Vanderkelen et al., 2021), the implementation of the Hanasaki et al. (2006) reservoir operation scheme in the vector-based global routing model mizuRoute (Mizukami et al., 2016; Vanderkelen et al., 2022) and implementation of different irrigation techniques including drip, sprinkler, paddy, and flood (Yao et al., 2022). At the time of this paper, the latter two developments are not yet available for usage in the release version of the CESM2, but are in the process of being integrated.

## 2.2 New sectoral water use module

The primary focus of our module development is to accurately depict the withdrawal and consumption of water across a variety of sectors. We define withdrawal as the gross amount of water removed from a water source for use in a particular sector. Sectoral water consumption, on the other hand, is the portion of water withdrawn that is actually consumed and not returned to the water source. It includes water that is lost through evapotranspiration, incorporated into products or crops, or otherwise not returned to the immediate water environment. This is achieved using a data-driven approach. The new module is designed to accept monthly expected water withdrawal and consumption data from non-prognostic sectors (all sectors excluding irrigation; Fig. 1). Expected daily water abstractions are then calculated within the land component of the model, CLM5, at grid-cell level (Fig. A1), by assuming a uniform distribution for all days within one month. To satisfy the water demand of each sector, water is provided from the river network, facilitated by a two-way coupling with the MOSART routing model. An existing coupling module between the land and routing components existed already for irrigation purposes. Its functionality was therefore adapted and extended to support the newly added sectors for both withdrawal and return flow fluxes.

During the coupling process, each CLM5 grid cell sends through the coupler to MOSART the information about how much water should be withdrawn and how much should be recycled back for each sector. The difference between the withdrawal and recycled part is the sectoral consumption, which is the net water amount which is transported from the river system to the land component. The CLM5 and MOSART spatial organization is different in this study, with CLM5 running on a 0.9x1.25° grid, while MOSART on a 0.5x0.5° grid. This needs to be taken into account when passing sectoral fluxes or water storage information from one model to the other during the coupling process.

In CLM, spatial land surface heterogeneity is represented through a nested subgrid hierarchy (Fig. A1) (Lawrence et al., 2019). Each grid cell contains multiple land units, columns, plant functional types (PFTs), and crop functional types (CFTs, if crop option is on). Land units, capturing the broadest patterns, include glacier, lake, urban, vegetated, and crop. Urban units are further divided into density classes. Columns represent variability within a land unit, such as different soil and snow states. Vegetated units may have multiple columns for soil profiles, while managed vegetation units have irrigated and non-irrigated columns. Columns have up to 25 layers for ground and 10 for snow, which allows solving for water storage and snow dynamics. The PFTs and CFTs corresponding to the third subgrid level, referred to as patches, represent various trees, shrubs, grass and crops covers that populate the given region. The patch level is intended to capture the biogeophysical and biogeochemical differences between broad categories of plants in terms of their functional characteristics (Lawrence et al., 2019). While the subgrid heterogeneity is captured by the model in the sense of realistic fractions of different land units, PFTs and CFTs, their

exact relative location is not represented. The calculations are done individually over each column and the outputs are then aggregated at grid-cell level before exchanging information with the coupler.

MOSART, which operates on its own grid, divides each spatial unit, such as a lat/lon grid cell or a watershed, into three hydrological categories (Fig. 1 from Li et al. (2013)): hillslopes that convert both surface and subsurface runoff into tributaries, tributaries that discharge into a single main channel, and the main channel that connects the local spatial unit with upstream/downstream units through the river network (Li et al., 2013).

The information about how much water is potentially available for sectoral use is provided by the coupler to the CLM5 model at grid-cell level by calculating the corresponding total river network storage in the MOSART model. This includes only the liquid water from the rivers, excluding iced river water or the water stored directly in the soils, which are not used to meet sectoral demands. Conversely, to send the information concerning each sector withdrawal and return flow to the MOSART model, a new module was implemented in the coupler codebase, that divides the sectoral fluxes across all main channels within the corresponding CLM5 grid cell in accordance to their relative weight in current water storage capacity. In this way, the main channels with larger current water storage will experience higher sectoral use than smaller capacity channels. For example, if MOSART has two active main channels within a CLM5 grid cell with a total water storage VOLR (variable name in the model), with the larger channel containing 80% of VOLR, and the smaller channel the remaining 20%, then the sectoral fluxes from the CLM5 grid cell will be distributed across the two available channels in the same proportion (i.e., 80%/20%). The same approach was originally implemented for the irrigation without the return flow part. The new module is therefore a generalization for the remaining sectors.

To simulate idealized scenarios to diagnose instances of water scarcity, the new module implements a basic sectoral priority algorithm for situations when water availability is inadequate to meet the full sectoral demand. Under this system, when water is scarce, it is allocated to sectors in the following priority order: domestic, livestock, thermoelectric, manufacturing, mining, and irrigation. Similar sectoral priority orders have been implemented in some Global Hydrological Models (GHMs; e.g., H08, Hanasaki et al. (2018) and VIC-5, Droppers et al. (2020)). This order reflects a general premise that priority should be given to high value-added products in resource allocation. Municipal, industrial, and agricultural water use intensities per value added are estimated at 0.012, 0.063, and $2.2 \times 10^6$ $m^3$ per $10^6$ USD, respectively (Hanasaki et al., 2018). This highlights that sectors such as municipal and industrial services provide higher economic returns per unit of water used compared to agriculture (Hanasaki et al., 2018).

It is at this stage that a distinction is made between expected and actual fluxes. Expected withdrawal or consumption is based on the input data estimates, while actual withdrawal or consumption represent the fluxes which are computed within the new module after water availability and sectoral competition are accounted for. Here we should mention that the irrigation and the new sectoral water use modules are kept separated within CLM, and the abstraction procedures are activated at different stages within the model driving loop. To achieve the connection of the two modules within the sectoral competition algorithm, we activate the abstractions for sectoral use before irrigation is treated, and then update the amount of available water perceived by the irrigation module by subtracting the withdrawal which was already done for the other sectors.

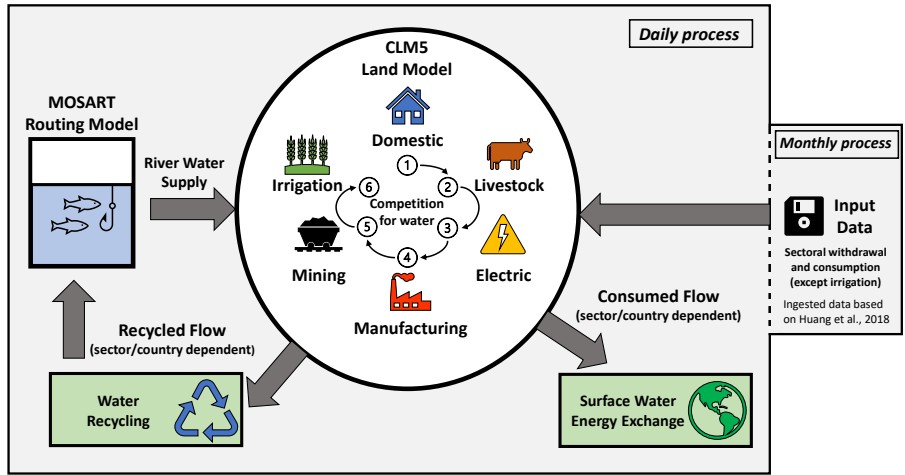

**Figure 1.** Schematic depiction of the new sectoral water use module implementation in the Community Land Model (CLM5) and the Model for Scale Adaptive River Transport (MOSART).

After water is allocated to each sector, the new module accounts for the return flow and consumptive use of the water (Fig. 1). A portion of the utilized water is recycled, meaning it is returned directly to the river network via the MOSART routing model following the coupling mechanism described previously. The return flow is computed for each sector by subtracting the actual consumption from the actual withdrawal. How much is returned in comparison to the withdrawal depends on the sector and country (Huang et al., 2018). For example, thermoelectric water use has the highest recycling rates, as there are few associated evaporative losses during the power plant's cooling process, while livestock has the lowest (Fig. C2). From a water availability perspective, consumed water is considered "lost", and in this module it is applied to the surface of soil columns covered with natural vegetation in the CLM5 land model. It is important to mention here that while the sectoral demand for non-irrigative sectors is generated at the grid-cell level, the consumption (the net transport of water from the river to the land model) is distributed at subgrid level, on the natural vegetation land unit (Fig. A1). This is done to not interfere with the cropland and urban land units, which have their own soil columns. Thus, sectoral consumption contributes to the surface water-energy balance primarily through evaporation, but also transpiration, infiltration, and runoff processes (Fig. A2). Assigning the sectoral consumption flux directly to the evaporation flux, as done in GHMs, is not a suitable option for CLM5. Owing to the strict requirements on water and energy conservation in CLM5 for coupled applications, transforming the consumed water into evaporation would require updating the surface energy partitioning accordingly. However, by applying the consumption flux on surface soil in naturally vegetated areas, this water can evaporate or not depending on energy availability, contributing to the surface water-energy exchange. It should be mentioned here that the total consumed flux is not applied on surface soil all at once, but dribbled out evenly during the modelled day. This is in contrast to irrigation, where the total withdrawal is distributed uniformly over a period of 4 hours starting with 6AM local time.

## 2.3 Input data

The new module for sectoral abstractions relies on input data for all sectors, except irrigation. Data on sectoral withdrawal and consumption is sourced from Huang et al. (2018). This dataset represents a historical reconstruction, which is generated by combining the US Geological Survey (USGS) and the Liu et al. (2016) improved Food and Agriculture Organization of the United Nations (FAO) AQUASTAT dataset on sectoral water use. It covers the period 1971–2010, and it is available on a regular 0.5x0.5° grid and monthly frequency. The represented sectors are domestic, livestock, thermoelectric, manufacturing, and mining. Irrigation is also represented, but we ignore it here, as we use the CLM's built-in irrigation module.

Domestic water withdrawal encompasses the use of water for various indoor household activities, including drinking, food preparation, bathing, laundry, dishwashing, and toilet flushing. It also includes outdoor uses like watering lawns and gardens, as well as water use by public sector and service industry. Electricity water withdrawal refers to the water used for cooling thermoelectric and nuclear power plants. Water withdrawal for mining is utilized in the extraction of minerals, which can be solids (like coal), liquids, or gases (such as natural gas). In manufacturing, water withdrawal serves multiple purposes including fabricating, processing, washing, cooling, or transporting products, incorporating water into products, and meeting the sanitation needs within the manufacturing facilities. These sectoral water withdrawal categories are consistent with the work of Liu et al. (2016), and described in Huang et al. (2018).

To get the final monthly gridded dataset, Huang et al. (2018) used a three-step approach, involving spatially downscaling the original country (or state) level data to the 0.5x0.5° grid level, followed by linear interpolation on the individual grid cells' time-series to get annual sectoral withdrawal from the 5-year interval from the reports, and ultimately, using a sector dependent temporal downscaling procedure to go from annual to monthly frequency.

For spatial downscaling Huang et al. (2018) used global population density maps from History Database of the Global Environment (HYDE; 1970-1980) and Gridded Population of the World (GPW; 1990-2010) for the domestic, thermoelectric, manufacturing and mining sectors, while using 2005 FAO global livestock density maps for the livestock sector, following the approach of Hejazi et al. (2014). A uniform distribution is adopted by Huang et al. (2018) for the temporal downscaling of water withdrawal of livestock, mining and manufacturing. For the domestic sector, a temporal downscaling based on the approach of Wada et al. (2011b) is used, where a modulating function is applied based on each grid cell's historical temperature ranges and a region-dependent amplitude parameter $R$ (Huang et al., 2018). Finally, the thermoelectric water withdrawal is temporally downscaled using the assumption that thermoelectric water use is proportional to the generated electricity, which is then estimated using heating degree-days (HDD) and cooling degree-days (CDD) as proxies (Voisin et al., 2013; Hejazi et al., 2015). The temporal downscaling algorithms, for both domestic and thermoelectric sectors, were validated and calibrated by Huang et al. (2018) based on existing observations.

The main idea of the Huang et al. (2018) dataset is to represent a reference for historical water use by being derived as much as possible from existing observation/reported data. While the usage of such reconstruction is of interest for historical applications, the new module for sectoral abstractions can accommodate alternative datasets for both historical and future

periods, which may be interesting for exploring the uncertainties related to the modelling of human water use and projecting its impact on water scarcity (Wada et al., 2016).

## 2.4 Understanding the algorithm through a hypothetical case study

To gain a comprehensive understanding of the sectoral water use module, we explore a hypothetical single-grid cell case study (Fig. 2). In this scenario, we track the dynamics between expected and actual withdrawal during a Northern Hemisphere summer season (JJA) for a hypothetical grid cell.

As detailed in Sect. 2.2, the model ingests gridded expected withdrawal and consumption data on a monthly basis (Fig. 1). From this monthly expected withdrawal, assuming uniform distribution, we compute the daily expected sectoral abstraction, which remains constant throughout a given month (Fig. 2a).

Certain sectors may exhibit a sudden increase or decrease in the expected withdrawal amount at the onset of a new month. This factor is influenced by the sector and the assumptions which went into the input data. For instance, for the domestic and thermoelectric sectors, the seasonality of the withdrawal in Huang et al. (2018) is modelled by executing temporal downscaling on the annual amounts using monthly surface temperature, as well as heating and cooling days as proxies. Conversely, other sectors may show no seasonality (Fig. 2c) due to the absence of strong dependencies (e.g., manufacturing), or because known dependencies such as livestock increased water requirements during heatwaves (Steinfeld et al., 2006) are not represented in the input data (Huang et al., 2018).

The spatial patterns of water use should also be considered, as they emerge from the utilization of various proxies for spatial downscaling. Some examples include population densities for domestic (Wada et al., 2011a; Hanasaki et al., 2018), nighttime light intensity for industrial (Droppers et al., 2020), density maps for livestock (Khan et al., 2023), power plant locations for thermoelectric (Flörke et al., 2013), and irrigation areas (Burek et al., 2020; Müller Schmied et al., 2021). The selection of proxies used for downscaling, the sectors modelled, and spatial resolution all influence the mix of sectors that may be represented in a given grid cell. In our hypothetical case study, mining and irrigation are not represented because we assume that no abstraction happens for these sectors in our hypothetical grid cell (Fig. 2f).

As explained in Sect. 2.2, CLM5 and MOSART can exchange information on local water availability at the beginning of each day, and adjust the actual sectoral abstractions while considering sectoral priority. In our example, we assume that a small local water deficit occurs in June, satisfying high-priority sectors like domestic and livestock fully, while the thermoelectric sector experiences a supply gap (Fig. 2d). While the sector higher in priority (thermoelectric) is not satisfied fully, no abstraction is happening for the sectors lower in priority (manufacturing; Fig. 2e).

Similarly, we suppose a larger water deficit in July–August, affecting the domestic demand directly (Fig. 2b). As a consequence, the other sectors only recover when the sector higher in priority is fully satisfied. While the scarcity and recovery processes are represented in this example with linear trends, these patterns are noisier in the model, following the day-to-day water balance dominated by precipitation, evapotranspiration, and runoff processes.

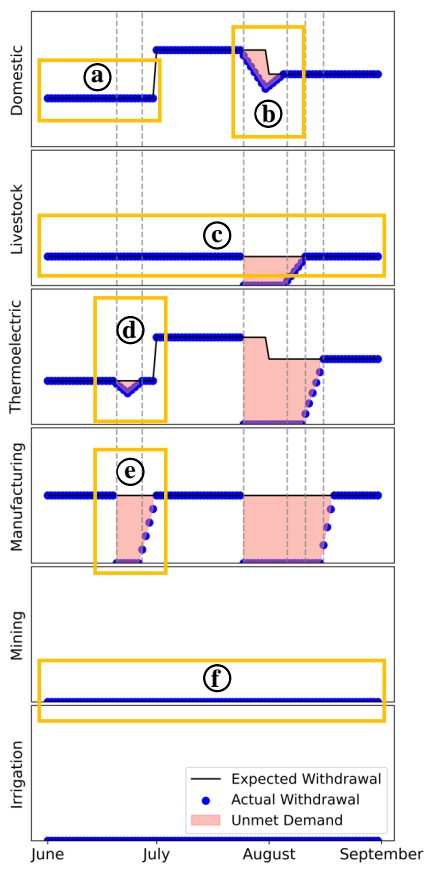

**Figure 2.** Illustration of potential outputs from the sectoral water use module for a hypothetical single-grid cell case study. The sectors are arranged by order of priority from top to bottom. The figure elucidates the interplay between expected (input based) and actual (taking into account limited availability) sectoral water withdrawals under both normal and water stress conditions. Factors such as river water availability and inter-sector competition, as previously outlined (Fig. 1), are integral to this model's function and results. Each labelled box represents a feature of the algorithm/model, which is discussed in Sect. 2.4. In addition, the interrupted vertical lines help identify the moments when water scarcity begins for a given sector and how the supply gradually recovers by order of priority.

## 2.5 Experiment set-up

CLM5 simulations are conducted for the period between 1971 and 2010, with the first 2 years excluded for spin-up (the analysis
period thus being from 1973–2010). Two simulations were conducted: a control simulation without sectoral abstractions and no irrigation, referred to as CTRL, and another simulation with complete sectoral water abstractions, including irrigation and the five new sectors, referred to as SectorWater.

Both simulations used a scientifically validated configuration designed for land-only simulations (IHISTCLM50BgcCrop component set). This configuration captures the historical changes in climate, $CO_2$-levels, transient land use and land cover

change including cropland expansion, uses the Global Soil Wetness Project atmospheric forcing data set (GSWP3v1), models terrestrial biogeochemical cycles, and includes a prognostic crop model (Lawrence et al., 2019). The simulations were run with a horizontal resolution of 0.9x1.25° and a default 30-minute time step. While it would have been possible to run simulations at higher resolutions (e.g. 0.5x0.5°), we opted for the 0.9x1.25° grid because it is one of the two scientifically supported grids for the IHISTCLM50BgcCrop configuration. In future applications, where the focus extends beyond demonstrating the module's
capabilities, a higher resolution setup may be preferred to provide more detailed regional insights.

Here we showcase the new module capabilities using land-only simulations. While testing the module in fully coupled CESM simulations is outside the scope of this study, this development is fully compatible with such applications.

## 3 Results

### 3.1 Testing and validation

Since we rely on a data driven approach, and the input data we are using is an evaluated reconstruction of historical water use (Huang et al., 2018), our validation focuses on the reliability and robustness of the implementation itself.

Initial tests focus on ensuring data consistency by verifying the robustness of remapping of the original sectoral water use dataset from 0.5x0.5° to 0.9x1.25° on the CLM5 land mask. In general, the remapping procedure is found to be conservative (Fig. B1), with relative errors of about 1-2% explained by upscaling effects when passing to a lower resolution land mask (Fig.
B2).

To verify correct behaviour, we compared expected, and actual model based sectoral abstraction fluxes with the input data. We confirm that expected fluxes at the grid-cell level match the input data, while actual fluxes are always lower or equal to expected fluxes when grid cell river networks lack sufficient water (Fig. B3–B4). The sector priority algorithm was confirmed to function correctly, with no lower-priority abstractions occurring when higher-priority sectors are unsatisfied (Fig. B5).

Next, the successful coupling between the land and the routing models is verified (Sect. 2.2). It was confirmed that the fluxes from the land model match the fluxes from the routing model at both global and continental level (relative errors <1%; Taranu (2024a)), demonstrating that differences in resolution and spatial organization were taken well into account during both the coupling process and the two-way remapping of the sectoral abstraction fluxes from land to the river network. While maintaining the same spatial patterns, the map corresponding to MOSART output (Fig. 3b) has more details and sometimes
more pronounced local values. This is because MOSART operates at a higher resolution in this case (0.5x0.5° versus 0.9×1.25° for CLM5) and has a different spatial organization with main channels of different storage. As a consequence, the sectoral fluxes from the CLM5 grid cell may be remapped unequally between MOSART intersecting grid cells (see Sect. 2.2 for a detailed description of the coupling process).

Finally, CLM5 ensures mass and energy conservation by aborting the simulation when this is violated.

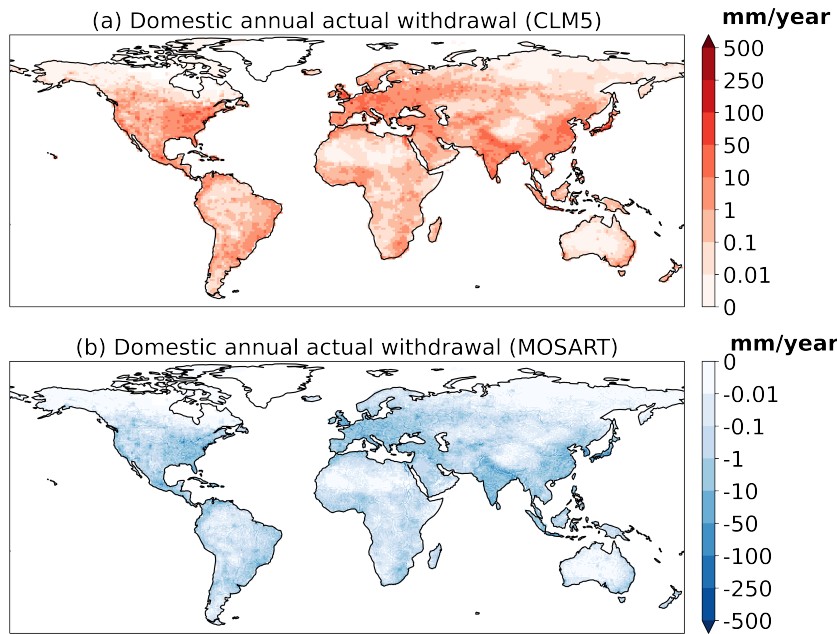

**Figure 3.** Actual domestic withdrawal for the year 2000 as outputted by the CLM5 (a) and MOSART (b) components. Sectoral withdrawal is a positive term for the land component, which gains water from consumption, and a negative term for the routing component, which loses water (Fig. 1).

## 3.2 Historical trends in global sectoral withdrawal

Over the period 1973–2010 (Fig. 4), sectoral water withdrawals show mostly a steady increase, followed by a slight decline towards 2010. While irrigation is consistently the largest contributor, other sectors account for a substantial share, especially looking at the actual withdrawal. By 2010, the cumulated non-irrigative expected sectoral withdrawal and consumption reached, 1157 and 171 km3/year respectively, which represent an increase of 315 km3/year or 36% for the period 1971–2010. The total global expected and actual water withdrawal increased by 110% and 43% respectively. Over the same period, in the SectorWater simulation, the expected (actual) irrigation withdrawal accounts for 79% (36%) of total water withdrawal, followed by domestic with 6% (19%), thermoelectric with 8% (23%), manufacturing with 5% (14%) and finally livestock and mining together at about 2% (8%).

Comparing the results on the relative importance of each sector with the results from Huang et al. (2018), reveals that the mean relative importance of irrigation is substantially higher in our case (79% versus 68%). To comprehend the findings of the SectorWater simulation, it is essential to consider two key factors. First, CLM5 computes irrigation water requirements prognostically, by taking into account for each day the soil moisture deficit which needs to be covered to satisfy crop water requirements. At the same time, in some regions, CLM5 struggles to supply enough water for irrigation using river water

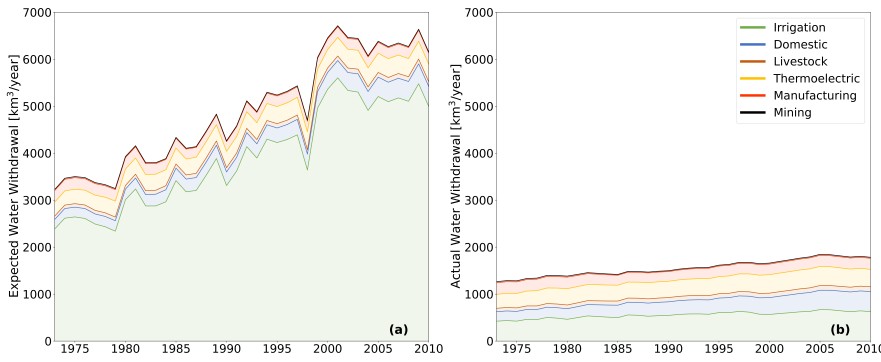

**Figure 4.** Annual global expected (a) and actual (b) sectoral withdrawal throughout the years 1973—2010. The actual withdrawal volumes can be lower than the expected ones because of limited river water availability and sectoral competition for limited resources (Fig. 1 and 2 for details). This figure was produced using the SectorWater experiment results. The trends and fluctuations in the expected irrigation withdrawal time-series are due to the changes in climate and historical cropland expansion, both captured in the SectorWater experiment (see Sect. 2.5 for experimental setup description)

.

alone (Fig. E6). When analysed globally based on the SectorWater experiment results, CLM5 supplies only 10-20% of what is
globally requested (Fig.C1.d), compared to >96% for all the other sectors. But this difference in supply should be interpreted cautiously. Due to the lack in water supply in certain areas, soil moisture levels are frequently below optimal levels. This results in a consistent water deficit for crops, leading to a greater reliance on irrigation. Consequently, this increases the overall annual expected irrigation withdrawals globally, which are likely exaggerated in our experiment.

     In fact, when only irrigation is considered and no limit on water supply is imposed, CLM5 underestimates the total irrigation
withdrawal in comparison to other models or AQUASTAT estimates. For example, in CLM5 it is possible to overcome the limitation of river water availability for irrigation supply through a negative subsurface runoff term in the MOSART model, which is then compensated with ocean water. This is an unrealistic configuration, but at the moment it is required to close the water balance when supplying the full irrigation requirements. It is also interesting to use when studying the land-atmosphere interactions and feedbacks of irrigation when the total withdrawal rather than the impact on discharge and runoff is important
(Thiery et al., 2017; Chen and Dirmeyer, 2019; De Hertog et al., 2023). In this configuration, and using a similar version of the model as in our study, Yao et al. (2022) found that the mean expected global irrigation water withdrawal for the period 1981-2015 is about 910 $km^3$/year. This is largely below the range given by the other models, of about 2000-4000 $km^3$/year for the same period (McDermid et al., 2023), and significantly lower than the expected irrigation withdrawal in our SectorWater experiment. Using the Yao et al. (2022) estimates for expected withdrawal, it seems that using river water alone could satisfy
up to 50% (vs. the 10-20% estimated based on our results) of the current CLM5 irrigation requirements (Fig. 4b). In this context, an analysis by sector of the fractions of unmet demands during the 1981–2010 period reveals that the global shortfall

in irrigation supply is not due to competition with recently introduced sectors. This conclusion is supported by numerous grid cells where only the irrigation sector exhibits undersupply (Fig. E1-E6).

This low estimate of global irrigation water withdrawal (910 km$^3$/year vs. 2000-4000 km$^3$/year) could be a consequence of the fact that the default irrigation technique used for all crops in CLM5 is very efficient, akin to drip irrigation. By introducing different irrigation techniques, including drip, sprinkler and flood, as well as a special parameterization for rice paddies, Yao et al. (2022) were able to reduce the bias in irrigation withdrawal compared to observations, increasing the total global irrigation withdrawal to about 3600 km3/year. Vanderkelen et al. (2022) also suggested that CLM5 is likely underestimating irrigation totals as well as irrigation seasonality, based on independent comparisons of reservoir inflows with observed values. Taking these results into account, and anticipating larger expected irrigation withdrawals in future versions of the CLM model in line with observations, and improved irrigation representation, it becomes critical that the water supply parameterization within CLM gets improved. This includes the implementation of abstractions from renewable and fossil groundwater, as well as from lakes and reservoirs.

## 3.3 Regional patterns in sectoral water use

Spatial patterns in sectoral water use vary significantly by region, influenced by climate, economic activities, and population distribution (Fig. 5). In areas experiencing precipitation deficits or with intensive agricultural activities, like the western U.S., eastern China, and the Indo-Gangetic Plain, irrigation demands are the highest (Jägermeyr et al., 2015; Huang et al., 2018). High sectoral water use aligns in general with the local density of the population, but there are also exceptions based on regional characteristics such as economic activities, climate, or water availability (e.g., dominance of eastern U.S. in thermoelectric water usage of the country).

Different socio-economic and climatic conditions lead to diverse water use profiles (see the map in Fig. 6). Agriculture-dominated water use is prevalent in many African, Asian, and South American countries, whereas industrialized nations in Europe and North America show a greater emphasis on industrial water use. Arid regions like the Middle East have higher domestic and agricultural water use due to increasing population and urbanization, while the high evapotranspiration rates and limited soil moisture makes irrigation essential for crop growth (World Bank, 2017).

From 1973 to 2010, water withdrawal trends changed, reflecting socio-economic progression, population dynamics, and changes in regional climate (see time-series subplots in Fig. 6).

In East and South Asia, there has been a notable rise in irrigation water use. This increase is likely attributed to warmer temperatures and expanding agricultural land for a growing population (Lombardozzi et al., 2020), both of which are captured by the SectorWater simulation (Sect. 2.5 for more details).

Most of the Global South regions experience a significant increase in their domestic and industrial water usage, likely a consequence of the increasing population, rapid urbanization and economic development (Wada et al., 2011a). At the same time, industrial water withdrawal in North America and Western Europe has stabilized or declined, potentially due to technological advances and economic shifts leading to higher water use efficiency (Flörke et al., 2013). Eastern Europe and Central Asia exhibit similar patterns, but likely influenced by post-Soviet transitions (Kummu et al., 2016).

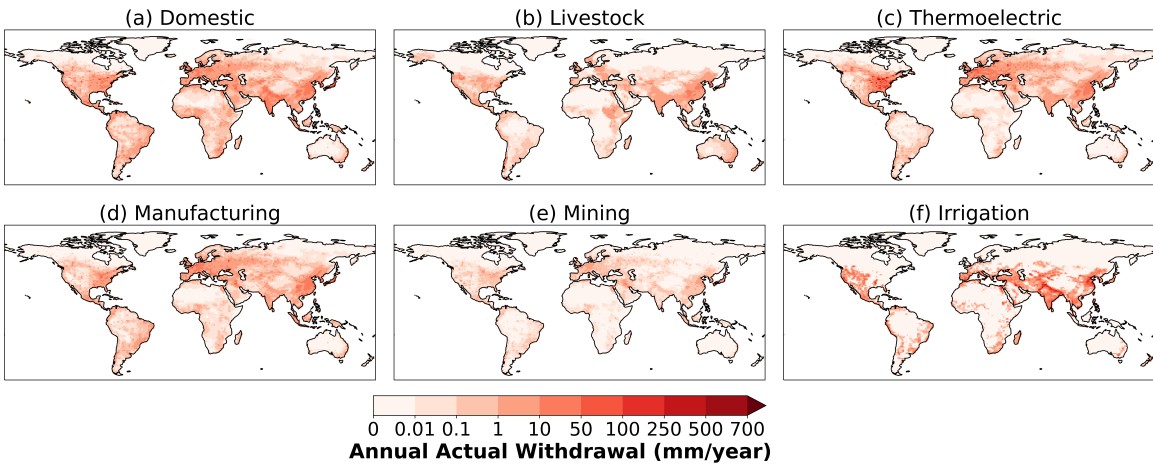

**Figure 5.** Spatial distribution of actual sectoral withdrawal for the year 2010 as outputted by the CLM5 land model in the SectorWater simulation. Note that the colour bar is non-linear.

Livestock water demand trends vary, with significant increases in North America, South America, and South Asia, following dietary changes and increasing global protein requirements (Rust, 2019). Some regions, however, have stabilized or decreased livestock water use, suggesting advancements in livestock production systems (Rust, 2019).

### 3.4 Impact on the climate

Irrigation withdrawal shows the strongest correlation with the regional climate differences when compared to other sectors' cumulative consumption (Fig. 8). A linear regression analysis, gives $R^2 \approx 0.9$ for irrigation withdrawal vs. $R^2 \approx 0.08$ for other sectors consumption in explaining the climatic variable differences between the CTRL and SectorWater experiments (Fig. 7 a, c, e). There are a few reasons why we don't find a significant effect for the non-irrigative sectoral consumption on the selected surface variables in our experiment. For example, for non-irrigative sectors: the cumulative consumption is quite small; it is
distributed over the year and the entire day (24h), and independently of soil moisture conditions; distributed over a larger area (natural vegetation column). As a consequence, the impact of non-irrigative sectors on the climate is insignificant at scales of a 100 km and above. For irrigation: the water is applied for parts of the year when the crops grow, and only when there is a moisture deficit; the irrigation window is short, applied over 4 hours at 6am local time; locally applied over the columns with irrigated crops; the total irrigation amounts being very large (e.g. maximum cumulative grid cell consumption for non-irrigative
sectors is only at the low end of irrigation grid cell consumption in Fig. 7). These conditions ensure that irrigation will have a significant impact on the local climate, since a significant amount of water is provided locally over a short period of time in a moment when evaporation and plant transpiration is limited by water availability.

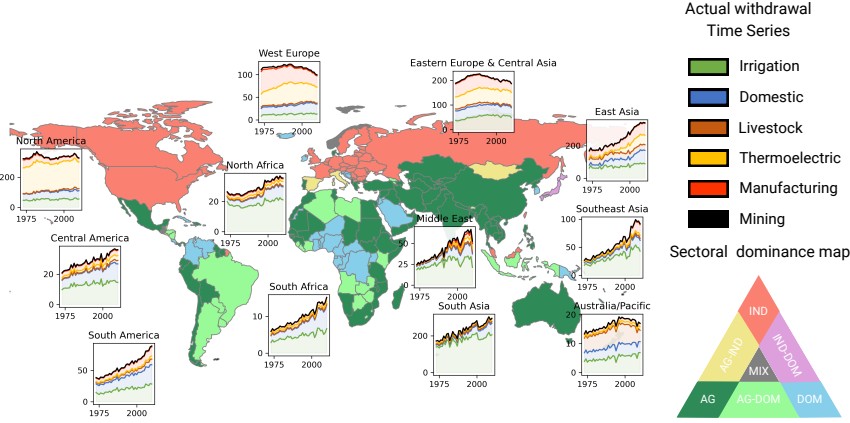

**Figure 6.** Countries' classification based on their dominant sectoral water use, as sourced from mean actual withdrawals for six sectors (irrigation, domestic, livestock, thermoelectric, manufacturing, and mining) between 1973 and 2010. For clarity, irrigation, and livestock data were combined under "Agriculture" (AG), while thermoelectric, manufacturing, and mining were aggregated as "Industry" (IND), and domestic is considered separately (DOM). A sector is dominant when it accounts for over 50% of total mean withdrawal throughout the observed period 1973—2010 (e.g., AG). A mixed usage is when no sector accounts for more than 40% or less than 30% of the mean total withdrawal, indicating an even distribution across sectors (MIX). Dual dominance is when leading sectors together represent more than 65% of total mean withdrawal, with neither falling below 25% (e.g., IND-DOM). The dominance map format is inspired by a similar map presented in the World Resources People and Ecosystems (2000). Accompanying the map are time series plots detailing actual withdrawals for the 12 major geographical regions.

These findings underscore irrigation's principal role as a climate forcing in relation to human water use (McDermid et al., 2023), with other sectors contribution being negligible at large scale. While not for their climate impact, incorporating other
sectors' abstractions into Earth System Models (ESMs) as suggested by Nazemi and Wheater (2015), remains important for evaluating water scarcity in present and future climates, as well as the changes of surface and groundwater storage resulting from sectoral withdrawal and consumption. For example, comparing the SectorWater and CTRL experiments, shows a significant decrease in mean annual streamflows for most major rivers (Fig. D1), as well as a decrease in total annual river discharge to ocean of about 300 km3/year by the year 2010 (Fig. D2).

**3.5 Water scarcity**

By visualizing the average number of days when the local supply of surface water was not sufficient to fully satisfy all sectoral demand (Fig. 9), we find that the results produced by the updated CLM model closely match known hotspots of water scarcity (Mekonnen and Hoekstra, 2016; Kummu et al., 2016; Liu et al., 2017).

Analysing model results, we find that the largest regional water scarcity is found in the Middle East. For example, by
405 aggregating results at sub-national level (Fig. E7), we find that the municipality of Al Khor, Qatar, experiences water scarce

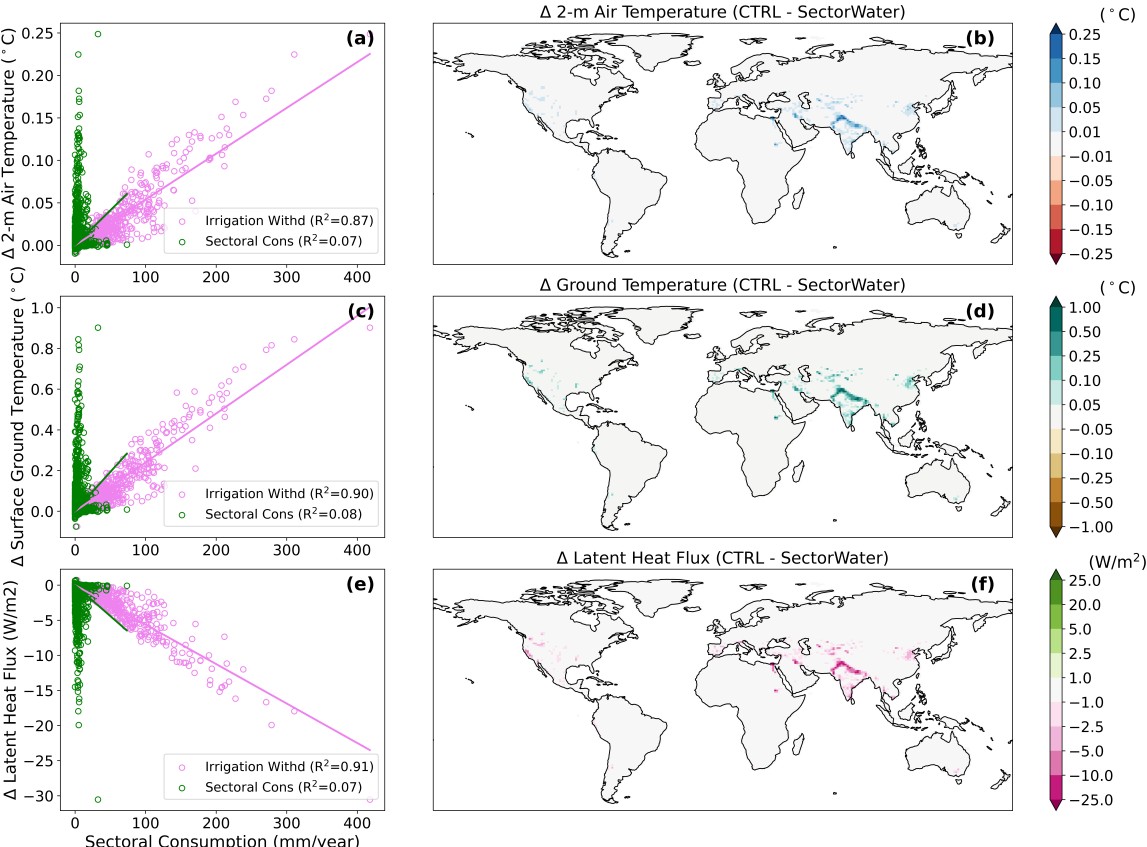

**Figure 7.** (b, d, f) Mean difference maps for 2-m air temperature, ground temperature and latent heat flux between the CTRL and SectorWater experiments. The maps are generated by calculating the difference between each month for the given variable between the years 1973—2010, and then computing the average over all the differences. (a, c, e) Linear regressions between the difference maps for the given variable, and one of the predictors, the mean irrigation withdrawal or the mean sectoral consumption (see predictors maps in Fig. 8). Each point in the regression plots represents individual grid cells from the corresponding right-side difference map.

conditions for about 320 days per year on average, where local river water is not enough to fully satisfy its residents demands. Other known hotspots of water scarcity in the Middle East (World Bank, 2017) are well captured by the model, including Haifa in Israel (195 days), West Bank in Palestine (140 days), Dar'a in Syria (125 days), Jizan in Saudia Arabia, Sharjah, and Dubai in the United Arab Emirates (about 90 days), Amran and Raymah in Yemen (about 150 days). The reasons are multifaceted:
from climatic challenges, with prolonged droughts and erratic rainfall, but also increasing populations and standards of living accompanied by increasing domestic and agricultural demands (Fig. 6). To manage these challenges, the region heavily depends on fossil groundwater exploitation, leading to rapid groundwater depletion (World Bank, 2017; Bierkens and Wada, 2019). As

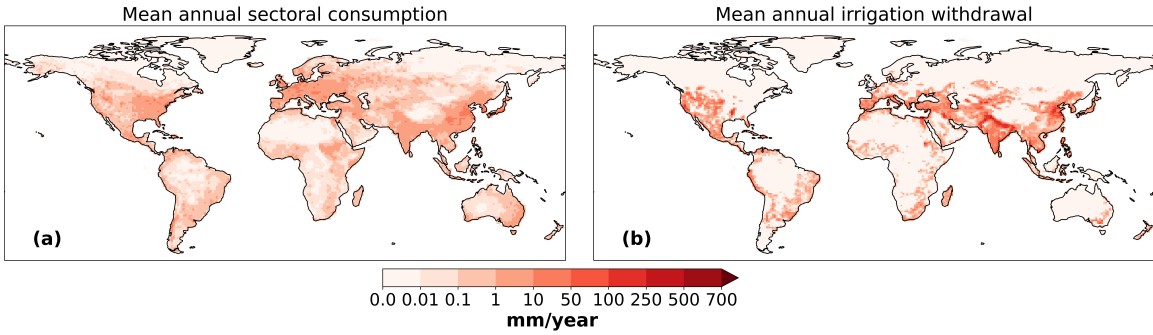

**Figure 8.** Spatial distribution of mean annual consumption for all sectors (excluding irrigation) alongside annual irrigation withdrawal, both used as predictors in the regression depicted in Fig. 7. It can be noted that irrigation withdrawal correlates much better with the surface climate variables in Fig. 7. While the irrigation withdrawal is in general much larger than non-irrigative sectoral consumption, this is especially the case for the grid cells where a climate modulating effect is observed (Fig. 7).

a result, the region makes substantial efforts to find alternative sources of water, such as desalination (Eke et al., 2020; Curto et al., 2021).

The Mediterranean, another region historically vulnerable to water stress, presents a similar narrative. Spain's Región de Murcia and Andalucia or Greece's Crete, all with over 100 days of unmet water demand in our simulations, are typical for the region, grappling with both climatic adversities and high agricultural demands. Climate change is expected to further intensify this issue, with projections indicating reduced rainfall and soil moisture in the region (Fig. TS.5 in IPCC AR6 Technical Summary: The Physical Science Basis, 2021).

Turning to Asia, the model results resonate with well-documented concerns. Regions in India, such as Karnataka and Maharashtra (about 120 days), and areas in China like Tianjin and Hebei (about 60 days), are well known for water scarcity. For example, previous studies showed water stress that ranges between 40-80% in Karnataka and Maharashtra region (Manju and Sagar, 2017), while water supply for the Beijing-Tianjin-Hebei urban agglomeration represents a significant challenge requiring trade-offs among economic development, environmental protection, and nexus risks in the adjacent regions (Cai et al., 2021). In Central Asia, the Sughd province in Tajikistan stands out with 100 days of unmet demands. This region is known

for high overall water scarcity, especially during peak irrigation seasons, leading to strong competition over water resources and a local practice of water rotation to ensure supply (Mukhamedova and Wegerich, 2018).

When it comes to North America, the region which stands out the most in our simulations is the state of California (60 days). With over 8.5 million acres of irrigated cropland, the state of California is vulnerable to the frequent and intensifying droughts,

and tends to rely on unsustainable groundwater exploitation (Mount and Hanak, 2019). This led to the adoption of the 2014 Sustainable Groundwater Management Act requiring pumpers to reach sustainability by the early 2040s. At the same time, the model seems to capture the continuing difficulties in Colorado (25 days). Between the period 1916–2014, the Upper Colorado River Basin experienced a 16.5% decline in naturalized streamflow, which is met by increasing population and sectoral demand

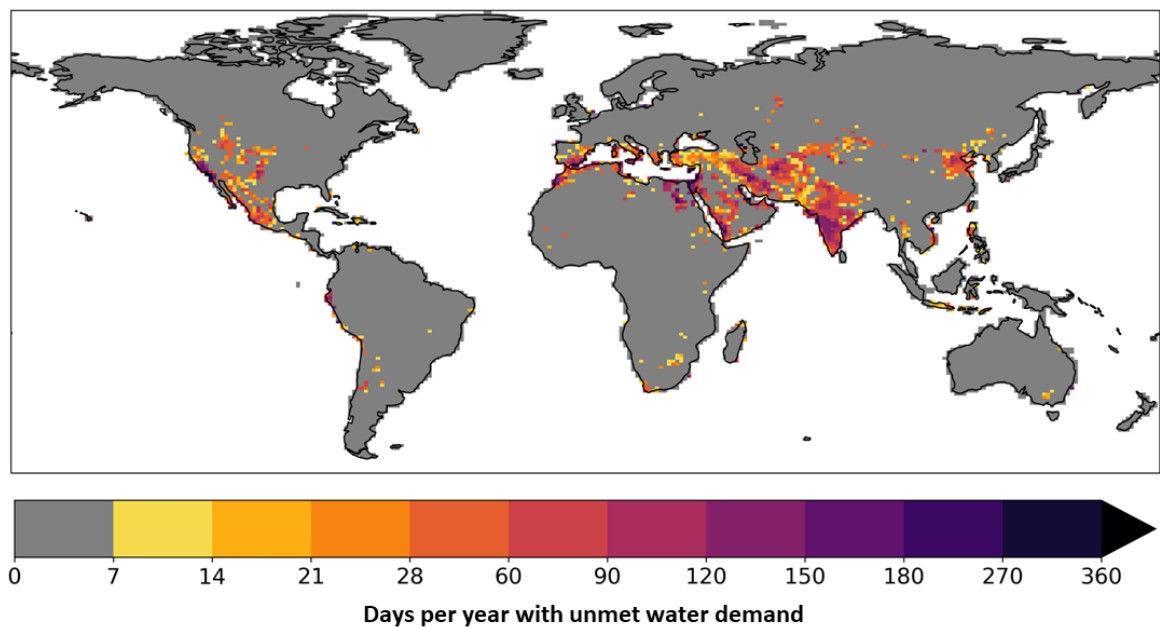

**Figure 9.** Average number of days per year, from 1973—2010, when modelled water supply was insufficient to meet the demand for all sectors. Note the non-linear colour bar.

often equal to the inflows (Xiao et al., 2018). For these reasons, and general low groundwater availability, the Colorado Basin is one of the basins with the highest projected economic impact uncertainty in the world in future climate scenarios (Dolan et al., 2021).

Other water scarcity hotspots which seem to be captured by the model are Western South America, Southern Africa, northwestern Africa, and Southeast Australia in line with previous research (Mancosu et al., 2015; Liu et al., 2017).

To understand better individual contributions, we analyse the same metric for individual sectors (Fig. 10). We notice that irrigation is by far the largest contributor to water scarcity, both by spatial extent and intensity. At the same time, we are observing significant number of days with unmet demands for the other sectors too in regions known to have difficulties with water supply. The regions which stand out for non-irrigative water scarcity are: Al Khor and Al Wakrah in Qatar, Toa Alta in Puerto Rico, Haifa in Israel, Al Marqab in Libya, Istanbul in Turkey, West Bank in Palestine, Al Isma'iliyah in Egypt, Makkah in Saudi Arabia, Tehran and Esfahan in Iran. As can be seen, the regions affected by non-irrigative water scarcity, in our analysis, are mostly located in the densely populated regions of the Middle East, Mediterranean, India, Western America, and Northern Africa regions. Due to the implemented prioritization order (see details in Sect. 2.4), we can see a cascading effect, with intensification of existing unmet demands and new regions being affected for each new sector down the priority order. These results are likely to change significantly depending on the allocation order, representing an important uncertainty

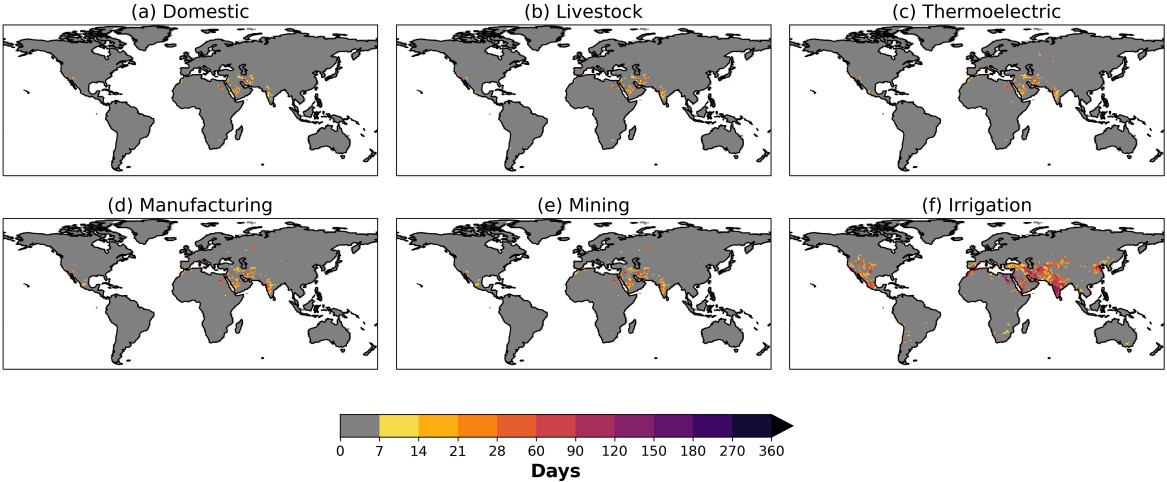

**Figure 10.** Average number of days per year, from 1973–2010, when modelled water supply was insufficient to meet the demand for individual sectors. Note the non-linear colour bar.

for individual sector's water scarcity assessments (Rathore et al., 2024). The complete analysis for each country at sub-national level can be found in the associated Data repository.

We find that for the period 1973–2010, a general trend of intensification of water scarcity is observed for most of the affected countries. This is expressed in an increasing fraction of the country struggling with sectoral water supply (e.g., India, Qatar, or Saudi Arabia in Fig. 11) or increasing number of days per year the affected regions are exposed to such conditions (Fig. 12). For some countries we see the first emergence of unmet sectoral demands (especially for non-irrigative sectors) only relatively recently (e.g. years 2000s for Brazil, China, Pakistan, Turkmenistan in Fig. 11). There are also situations, when unmet sectoral demands become less common (e.g. Russia in Fig. 11). In most cases, the observed long-term trends in the historical water scarcity can be explained by changes in countries sectoral demands (Fig. 13). With increasing sectoral water needs, many regions reached or are over their surface water availability limit (Wada, 2016). At the same time, there are also some exceptions which cannot be explained simply by changes in sectoral demands (e.g. the drop in domestic/livestock water supply in US between 1980s-1990s, or the drop between 1990s-2000s in Saudi Arabia all sectors supply). This indicates the existence of some other sources of water scarcity, such as long-term natural variability (Rodell et al., 2018). While a possible application of our module, in this study we do not aim at disentangling exactly the reasons for unmet demands between natural variability, climate change and changes in sectoral demands.

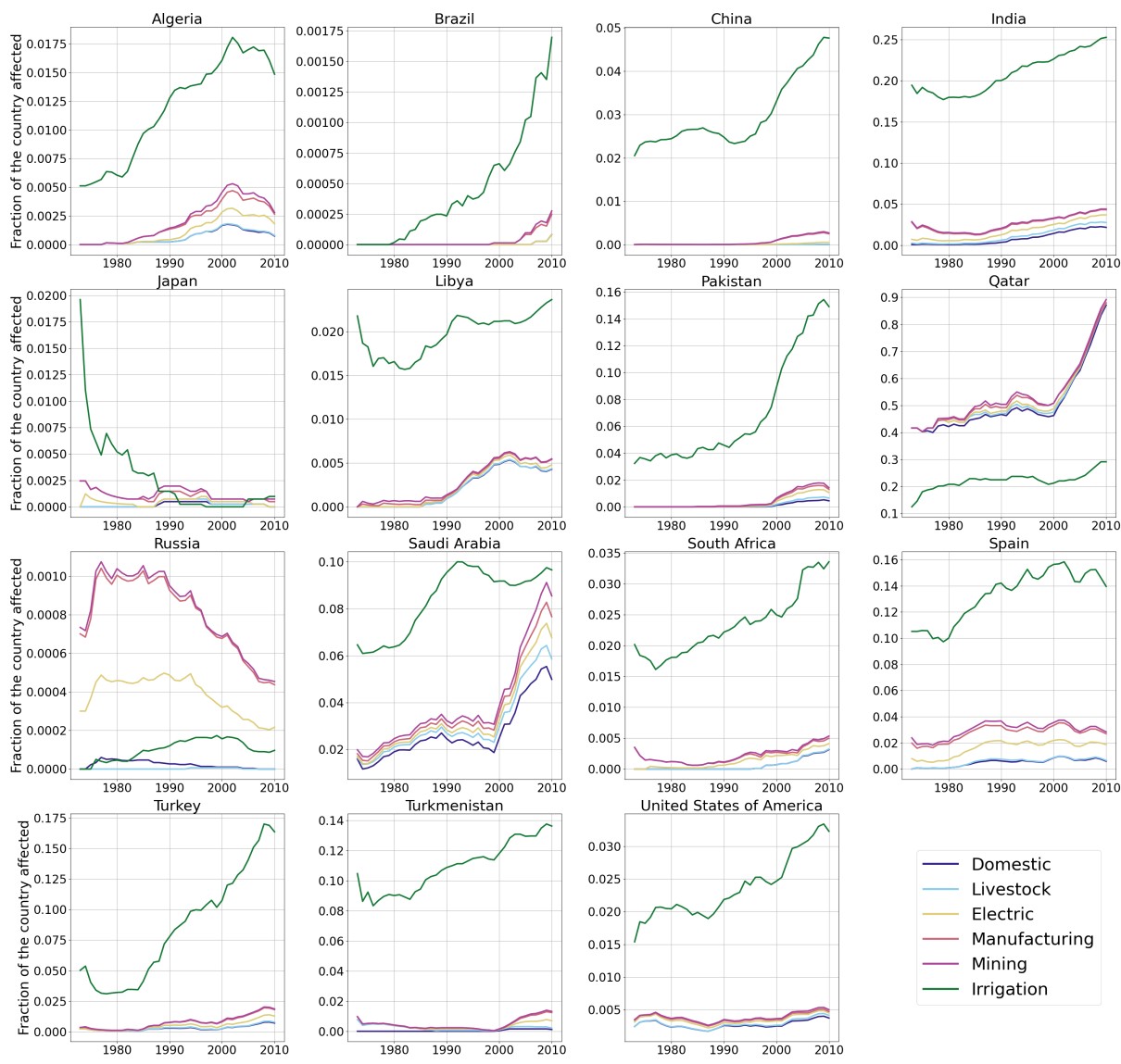

**Figure 11.** Fraction of the country with at least one day per year when modelled water supply was insufficient to meet the demand for individual sectors. The results are provided as a 10 years rolling average for better representation of long-term trends.

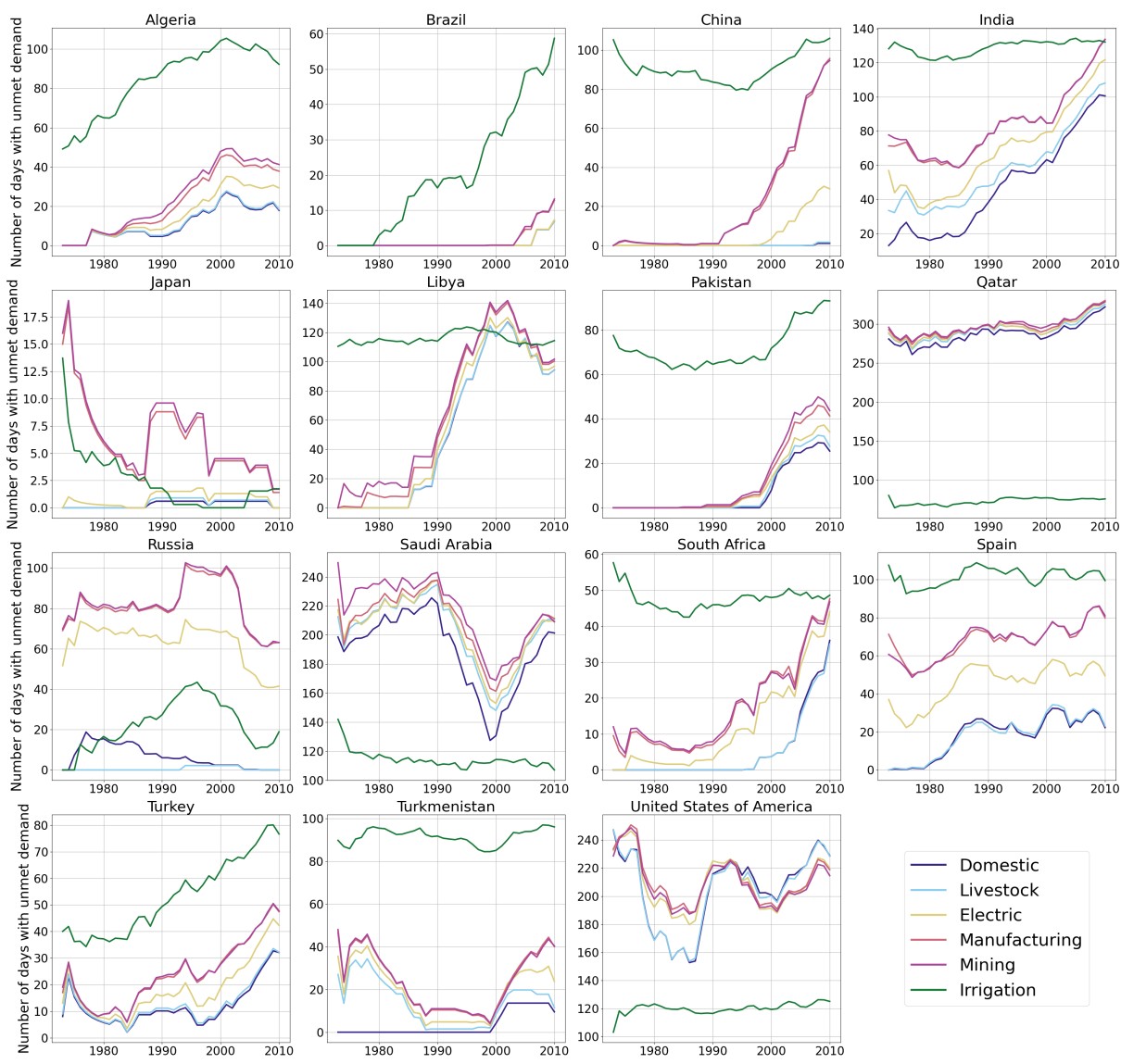

**Figure 12.** The average number of days per year, countries are exposed to unsatisfied sectoral demand. The average is calculated only using the grid cells which experience water scarcity conditions. The results are provided as a 10 years rolling average for better representation of long-term trends.

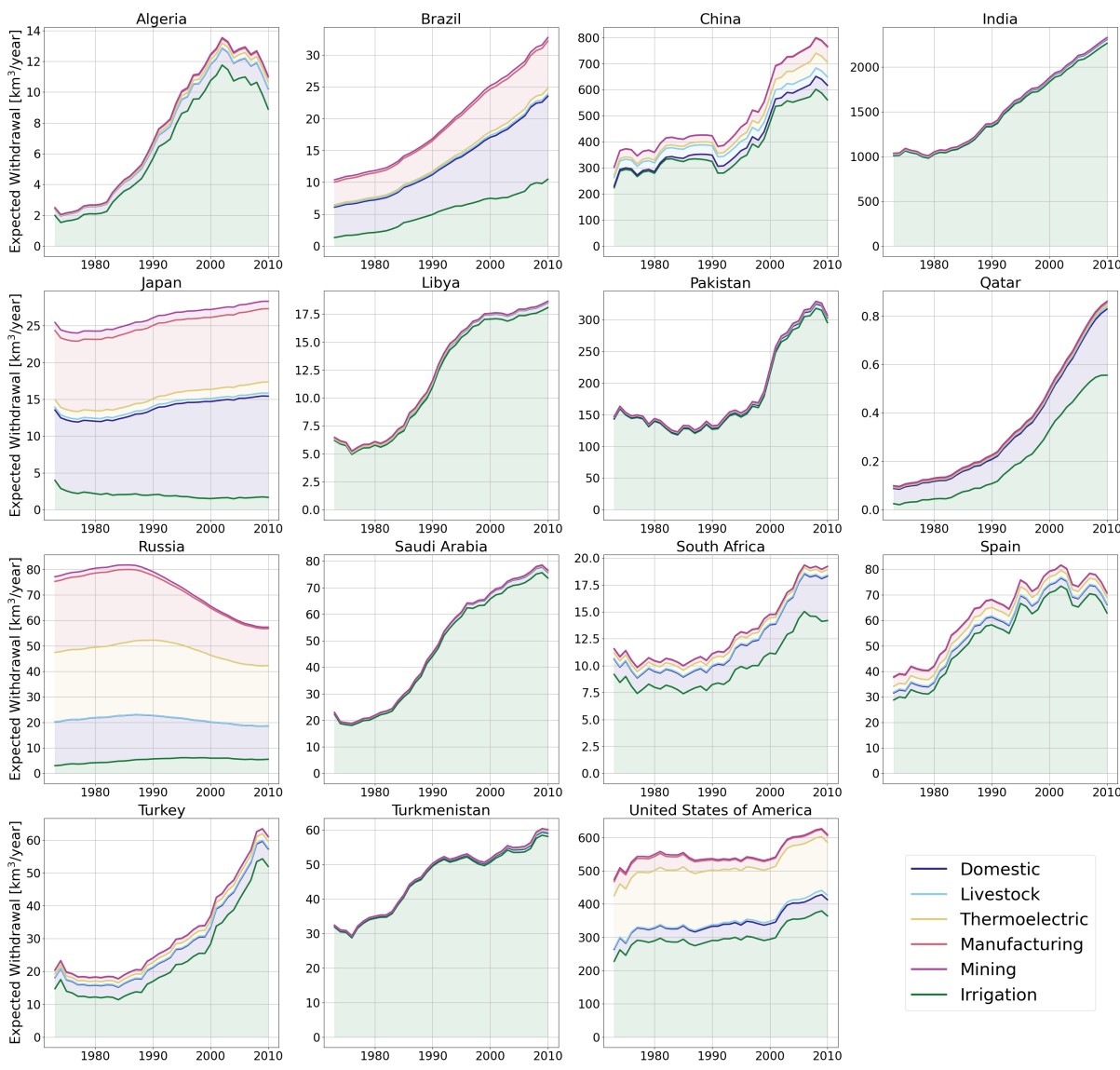

**Figure 13.** Countries sectoral withdrawal for the years 1973–2010. The results are provided as a 10 years rolling average for better representation of long-term trends.

## 4 Limitations and a way forward

While our additions to CLM5 represent a notable advancement in the representation of human water use in the Community Earth System Model (CESM), there are still some limitations and assumptions that should be acknowledged.

In the latest iteration of the CLM model, an alternative mechanism allows for the extraction of unmet water demand for irrigation from unconfined groundwater in addition to the supply from the rivers (confined aquifers are currently not supported by the model). This approach provides a more realistic depiction; however, a potential limitation is its exclusive reliance on
model-calculated renewable groundwater availability. Given the present model's omission of water abstractions from reservoirs and lakes, there's a likelihood of overestimating groundwater dependence in certain areas. Conversely, there is a potential for significant underestimation of groundwater abstractions in arid and semi-arid regions. These regions often experience minimal groundwater recharge (Bierkens and Wada, 2019), and the model currently does not account for fossil groundwater reserves. Nevertheless, comprehensively accounting for every source of sectoral water withdrawal is crucial for the valid application of
CLM5 in water scarcity assessments. Therefore, evaluating the efficacy of this new groundwater abstraction approach, along with its expansion to recently incorporated sectors, emerges as an imperative future effort.

In the context of groundwater abstractions, it is also important to consider how the partitioning between surface vs. ground-water dependence is implemented. The method currently available in CLM for irrigation is what can be called an implicit method, where the amount supplied from groundwater is based on what remains unsatisfied from surface water (rivers). The
480 advantage of following this approach is that it gives better estimates, especially in regions where significant groundwater pumping remain unreported (Wada, 2016). However, the implicit method may neglect physical, technological and socioeconomic limitations in groundwater use that exist in various countries (Wada, 2016). Alternative methods exist, which rely on national and subnational statistics to calculate for each sector the fraction of withdrawal satisfied by source (Döll et al., 2012). Such methods are more likely to capture regional/national patterns of groundwater use, but may be too conservative due to the prob-
485 lem of unreported usage and lack of reliable data for many countries (Döll et al., 2012; Wada, 2016). We think that a mixed approach, where the fractions of surface vs. groundwater usage per sector are given but not fixed, may be of interest. With increased quality and availability of remote sensing data, such as GRACE (Gravity Recovery and Climate Experiment), we can imagine using the fractions of surface vs. groundwater usage as a model calibration parameter to better constrain groundwater abstractions using the observed terrestrial water storage changes (TWS) (Anderson et al., 2015; Wada, 2016).

In addition to the implementation of more conventional sources of water supply including rivers, lakes, reservoirs, and groundwater, it may be important to also consider alternative sources such as desalination and treated wastewater (Van Vliet et al., 2021). A recent assessment showed that desalination capacities are increasing globally at an exponential rate, and in 2020 the annual production was at about 35 km$^3$/year (Jones et al., 2019). In addition, for the year 2015, wastewater was produced at a rate of about 360 km$^3$/year globally, from which only about 10% was intentionally re-used after treatment (Jones et al.,
2021). While these unconventional water sources are still two orders of magnitude below the global sectoral withdrawal, they are most often employed in water scarce regions (Jones et al., 2019, 2021), where they can significantly reduce severe water scarcity and people affected by it (Van Vliet et al., 2021).

Van Vliet et al. (2021) also underscore the significance of water quality in exacerbating water scarcity. The research reveals that 40% of the global population faces severe water scarcity when accounting for both water quantity and quality, compared to 30% when only quantity is considered. In the future, more GHMs/LSMs may incorporate water quality indicators (e.g., surface water temperature, salinity, organic pollution) in order to provide more precise information on both water quantity and quality. In comparison to GHMs, the LSMs possess the distinct advantage of having coupling capacity with other components (e.g., ocean, atmosphere) and, often, existing tracing modules (e.g., for tracing isotopes). As such, development of water quality indicators in LSMs may be easier and have a wider application range (e.g., tracing pollutants in the ocean after discharge). In this context, when developing the sectoral abstraction modules in other GHMs/LSMs, maintaining the withdrawal and return fluxes separately for each sector will be important. While more redundant and computationally more expensive, it allows for an easier implementation of a water quality module in the future by connecting the sector-dependent pollutants (e.g., temperature for thermoelectric, or N/P for livestock) to the return flow of each sector (Van Vliet et al., 2021). At the same time, water quality indicators will help calculate sector-dependent extra withdrawals for dilution to obtain acceptable water quality levels for each sector, which will improve water scarcity assessment capabilities of the models (Van Vliet et al., 2021).

The over-abstraction of surface water at the expense of environmental flow is another important aspect of water security that needs to be addressed in the GHMs/LSMs models development. Today, more than a quarter of the surface water consumption in South, West, and Central Asia, northeastern China, Spain, and Argentina is considered unsustainable or at the expense of the environment (Wada, 2016). Because of increasing human water use (Wada et al., 2016) and drier climate conditions (Trenberth, 2011), new hotspots of non-sustainable surface water use are emerging in the USA, Mexico, the Mediterranean, the Middle East, Northern, and Southern Africa (Wada, 2016).

At the moment, the way environmental flow requirements (EFRs) are treated in the CLM model is by limiting the amount which can be subtracted for sectoral use to 90% of the current river water availability. This approach allows avoiding the complete depletion of rivers, but it is likely severely overestimating the amount of water which can be abstracted for sectoral use. For example, global assessments have shown that on average, about 37% of annual discharge is required to sustain EFRs (Pastor et al., 2014). During low-flow periods, the EFRs are even larger and may need 46-71% of the available water (Pastor et al., 2014). In order to improve the capabilities of CLM to account for both human and environmental water needs, the implementation of variable flow based methods to estimate EFRs is recommended. These methods classify the flow regime into high, intermediate, and low-flow months, and take into account the intra-annual variability of flow conditions, thus providing better estimates of EFRs under a variety of flow regimes (Pastor et al., 2014).

In our model, sectoral water use priorities are currently fixed and follow the order (in decreasing priority) domestic, livestock, thermoelectric, manufacturing, mining, and irrigation. While a similar hierarchy was also implemented in some other models (Hanasaki et al., 2018; Droppers et al., 2020), in reality the priority may vary based on regional circumstances, weather, policies, or changing socio-economic conditions. For example, a recent study suggests that in many regions the domestic and irrigation sectors often receive higher priority than other sectors during periods of droughts, heat waves, and compound hot-dry extremes (Cárdenas Belleza et al., 2023). At the same time, regional exceptions are possible, highlighting the need for more flexible approaches in modelling sectoral competition in GHMs and LSMs models (Cárdenas Belleza et al., 2023). For

example, a recent study explored an alternative prioritization where agricultural demands are placed first (Rathore et al., 2024). This scenario resulted in around 30% increase in unmet demands for municipal (domestic) and industrial sectors for urban areas. If we were to replicate this experiment, similar results would be anticipated, as evidenced by the figures in Appendix D2-D6 where many grid cells experiencing unmet demand for irrigation, do not experience such scarcity for the other sectors. This further supports the development of more flexible prioritization schemes to study related uncertainty in unmet sectoral demands.

While more model development may be needed to represent relevant processes related to human-water interactions, another important aspect to consider is model evaluation and calibration for hydrological variables. The variables which are the most important for water availability modelling are precipitation, evapotranspiration, snowpack dynamics, glacial melt, soil moisture, surface runoff, river flow, and groundwater levels and recharge. For example, Vanderkelen et al. (2022) showed that while globally the runoff biases in CLM5 are very small (+0.077 mm/day), large regional biases exist. Aggregated at the level of a catchment, such biases can result in significant river discharge biases, limiting the model usability for water management purposes (Mizukami et al., 2021). Efforts are being made to solve this problem with targeted evaluation studies to understand hydrological parameter uncertainty in CLM5 (Yan et al., 2023). At the same time, more efficient and transparent objective calibration protocols to improve model performance for a given set of targets are being developed (Dagon et al., 2020; Cheng et al., 2023). Unfortunately, running large parameter perturbation ensembles for sensitivity testing and application of objective calibration protocols remains very expensive for LSMs/ESMs, and are usually done only for the release versions of the model. In the future when the model is calibrated, it could be interesting to expand our analysis by assessing the added value of the implementation of human water management on river flow and other relevant hydrological variables. For example, in the case of GHMs, it was found that considering human related impacts, including land-use-change, reservoir operations and water abstractions results in a general performance increase to represent streamflow and hydrological extremes (Veldkamp et al., 2018).

Finally, a key limitation of prescribing human-water use within ESMs, is the potential lack of temporal coherence between atmospheric conditions and sectoral water use. For instance, Huang et al. (2018) used the WATCH Forcing Data methodology to ERA-Interim reanalysis data for 1971-2010 (WFDEI, Weedon et al. (2014)) for temporal downscaling in the domestic and thermoelectric sectors, using gridded daily air temperature as a proxy. However, a land-atmosphere or fully coupled ESM simulation for the same period, initialized with historical accurate data, may generate atmospheric conditions diverging significantly from observed data due to the model's internal variability (Deser et al., 2012). Currently, there is no known method to fully reconcile these discrepancies.

As an interim measure, one approach is to prescribe sectoral water use data with no monthly temporal variation, effectively distributing it uniformly across the year. The main problem with this approach, is that it will greatly limit our ability to understand the impact of hot and dry extremes on water scarcity and sectoral competition. A more suitable solution would be the development of new algorithms for sectoral water use modelling that, similarly to irrigation, are prognostic, i.e., dependent on simulated environmental conditions, such as model-computed daily air temperature. In this case, the introduction of our new module offers a useful framework on which such development can be built.

## 5 Conclusions

The increasing global challenges surrounding water scarcity highlight the need for advanced modelling tools that can accurately capture human-water interactions. This study makes a contribution in this direction by implementing a data-driven sectoral abstraction module within the Community Earth System Model (CESM) framework. The enhanced model accounts for water abstractions in domestic, livestock, thermoelectric, manufacturing, mining, and irrigation sectors. It closes the water balance by integrating water abstractions from the land component with the supply and return flows from the river component. A basic sectoral competition algorithm was implemented to account for demand-supply dynamics when water availability is below the total demand. As a consequence, water scarcity dynamically emerges in our model, calculated daily as the gap between local demand and supply for each sector.

We validated the robustness of the implementation of the new sectoral abstractions module in CESM and conducted simulations for the period from 1971 to 2010. These simulations compared a scenario without sectoral water use to one with water use across all sectors. Results were used to analyse simulated historical global water withdrawal trends, regional variations in water use, the influence of sectoral water consumption on local climate, and the model's ability in identifying known water scarcity hotspots.

While irrigation is the largest user of water globally, we showed that in many regions other sectors, like domestic or industrial, dominate. This challenges the usual focus on irrigation in Earth System Models and points to the increasing importance of non-agricultural water demands in areas experiencing rapid population growth and socio-economic development. Through the implementation of all major water use sectors, it is now possible to study water scarcity in more details, by analysing sector specific unmet demands and impacts.

Our findings show that only irrigation has the potential to significantly affect local climates (for scales above a 100 km), while the effect of non-irrigative sectors is negligible. This might not be true at higher resolutions, especially if we would consider groundwater abstractions and land-atmosphere coupling. For example, Keune et al. (2018) study reveals that groundwater abstraction can significantly weaken the continental sink for atmospheric moisture by reducing soil moisture and altering surface energy fluxes. This reduction in soil moisture leads to decreased evapotranspiration, which in turn can diminish the local recycling of moisture back into the atmosphere. The diminished recycling can lead to reduced precipitation in some regions, thereby exacerbating local drought conditions. Furthermore, the weakening of the continental moisture sink due to groundwater depletion can have far-reaching implications for weather patterns and regional climate stability. While we find that the climatic impacts of other sectors like domestic and industrial water use are comparatively small, their inclusion in the model remains important for water scarcity assessment capabilities.

The model's simulations adeptly capture global hotspots of water scarcity identified in previous research on the topic (Mancosu et al., 2015; Liu et al., 2017). These results are promising and show the potential of CESM and its land component, the Community Land Model (CLM5), as a tool for future water scarcity assessments. In this regard, with the new sectoral water capability, the CLM5 model is well positioned for research on water use and scarcity, paralleling the capabilities of Global Hydrological Models (GHMs). This feature enables CLM to act as an impact model for the water sector, contributing to ini-

tiatives such as the Inter-Sectoral Impact Model Intercomparison Project (ISIMIP; Frieler et al. (2017)). Additionally, it paves the way for the incorporation of this sectoral water use data into Earth System Models (ESMs) simulations in a coupled mode. Although the direct feedbacks on climate from this additional sectoral water use are relatively minor, they play an important role in realistically modelling of other ESM variables, such as runoff and river flow.

*Code and data availability.* The model code with the new sectoral water use module, as well as the experiment setups, can be accessed with the following DOI: https://doi.org/10.5281/zenodo.10579224 (last access 29th January 2024). The data needed to reproduce this study can be accessed with the following DOI: https://doi.org/10.5281/zenodo.10518843 (last access 29th January 2024). The scripts used in this study are available at https://github.com/VUB-HYDR/2024_Taranu_etal_GMD, with the following DOI: https://doi.org/10.5281/zenodo.12675434 (last access 6th July 2024).

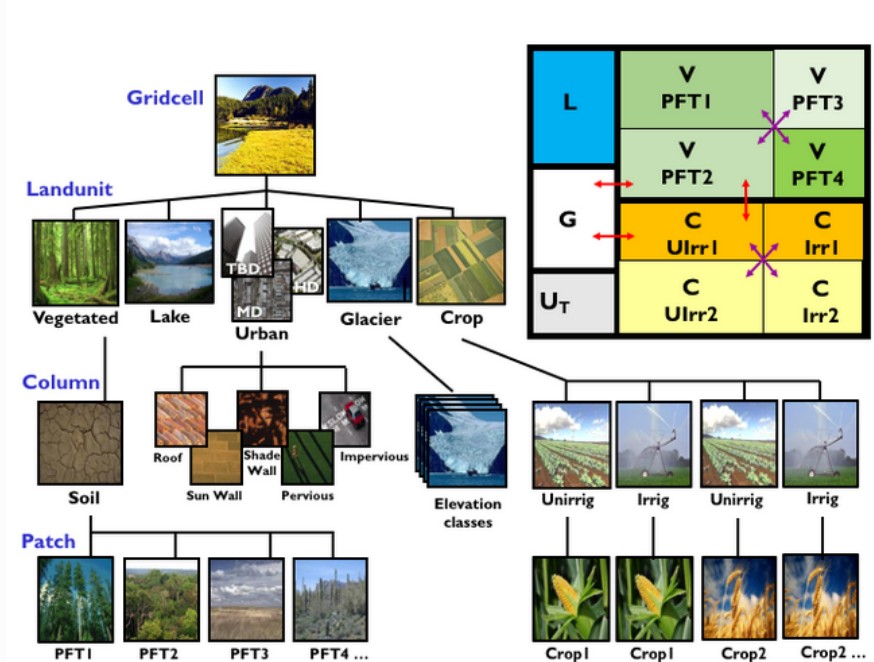

**Figure A1.** Standard configuration of the CLM5 subgrid hierarchy. Box in upper right shows hypothetical subgrid distribution for a single grid cell. Note that the crop land unit is only used when the model is run with the crop model active. TBD = tall building district; HD = high density; MD = medium density; G = glacier; L = lake; U = urban; C = crop; V = vegetated; PFT = plant functional type; Irr = irrigated; Rnfd = rainfed. Red arrows indicate allowed land unit transitions. Purple arrows indicate allowed patch-level transitions. Figure taken from Lawrence et al. (2019) and used with author permission.

## Appendix A: Modelling framework

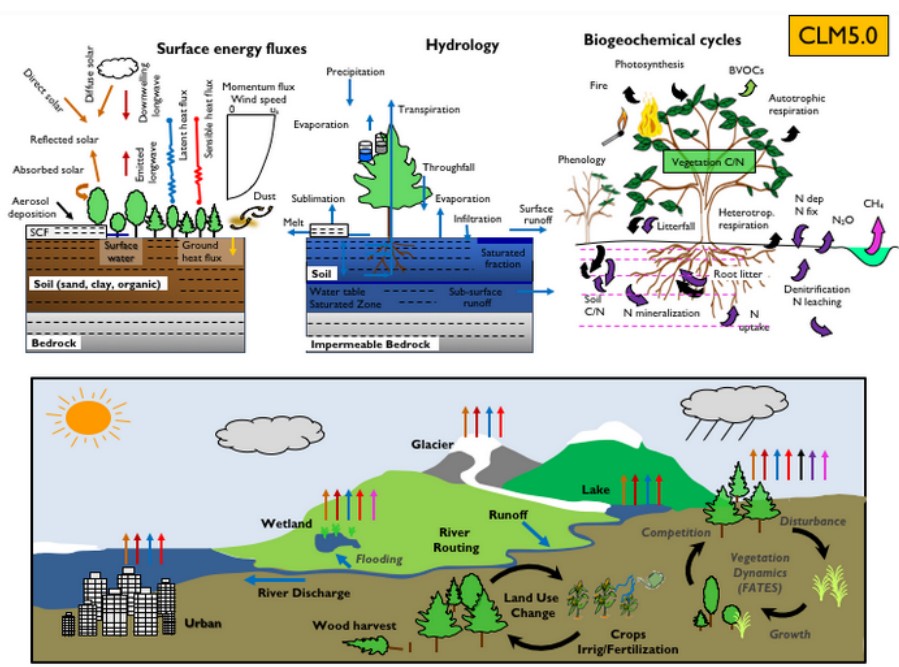

**Figure A2.** Schematic representation of primary processes and functionality in CLM5. SCF = snow cover fraction; BVOC = biogenic volatile organic compounds; C/N = carbon and nitrogen. For biogeochemical cycles, the black arrow denotes carbon flux, and the purple arrow denotes nitrogen flux. Note that not all soil levels are shown. Not all processes are depicted. Optional features that are not active in default configurations are italicized. Figure taken from Lawrence et al. (2019) and used with author permission.

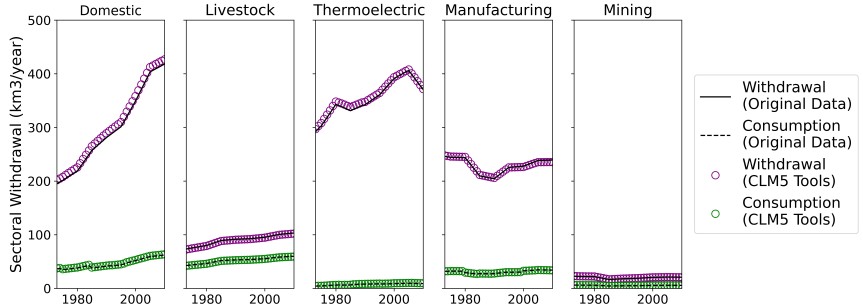

**Figure B1.** Comparison between global annual sectoral water withdrawal and consumption computed from the original dataset at 0.5x0.5° resolution (black lines) and the preprocessed dataset at 0.9x1.25° on the CLM5 land mask (colored circles). The remapping process was done using existing CLM5 tools modules, by extending support for sectoral water use datasets (Taranu, 2024b).

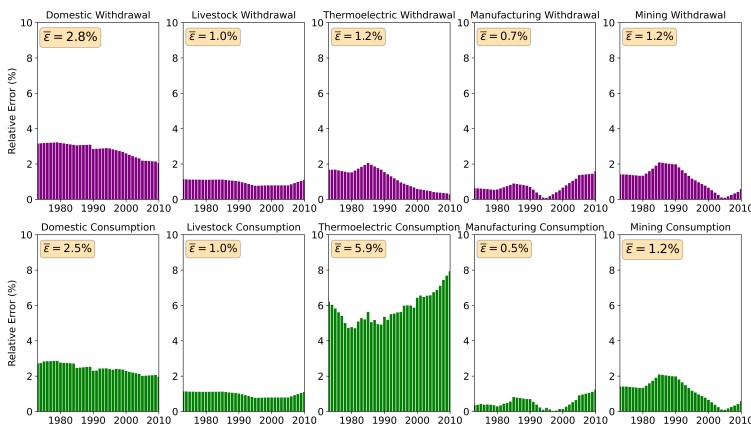

**Figure B2.** Relative errors between global annual sectoral water withdrawal and consumption computed from the original dataset at 0.5x0.5° resolution and the preprocessed dataset 0.9x1.25° on the CLM5 land mask (Fig. **??**).

## Appendix B: Sectoral use module validation

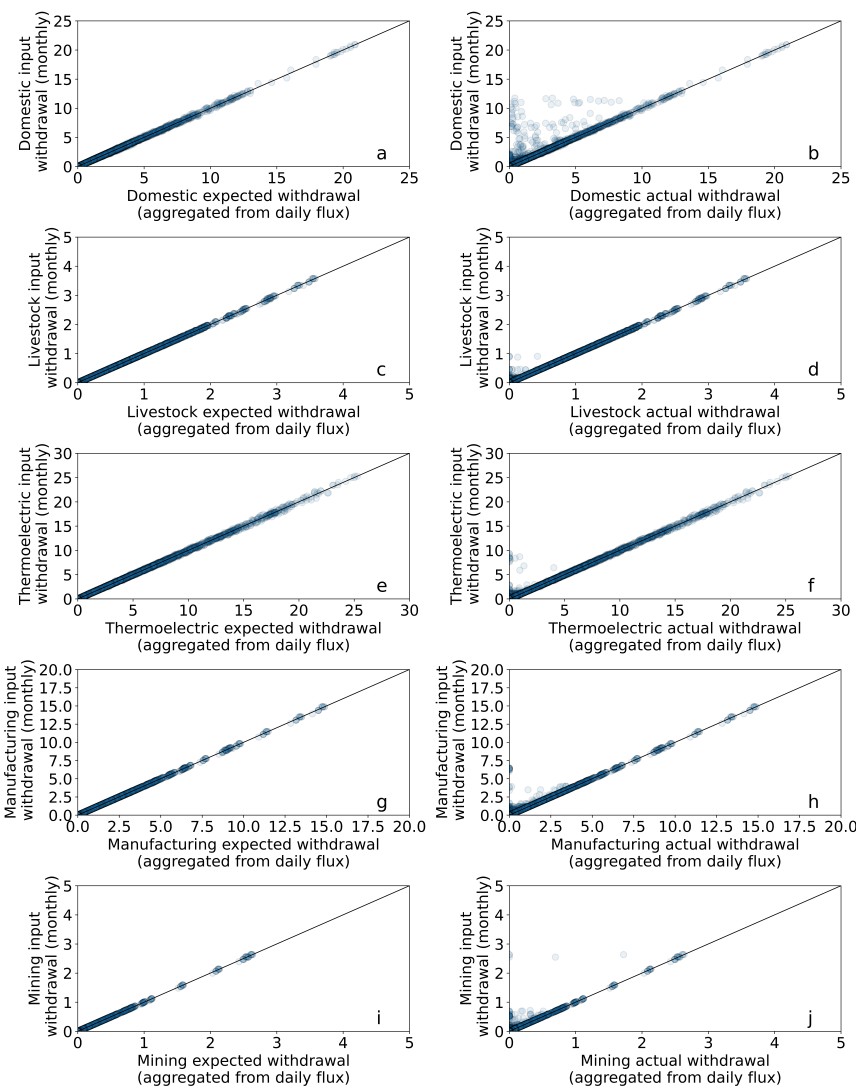

**Figure B3.** Comparison between expected monthly sectoral withdrawal values from input data, and monthly values aggregated from daily expected or actual withdrawal from model outputs. Each point represents the monthly value for a given grid cell. The points are plotted semi-transparently (alpha=0.5), therefore the more intense coloured parts simply indicate a larger concentration of values in that range. This plot was made by using the outputs for the year 2000 of the SectorWater experiment, and units for both axis are mm/month.

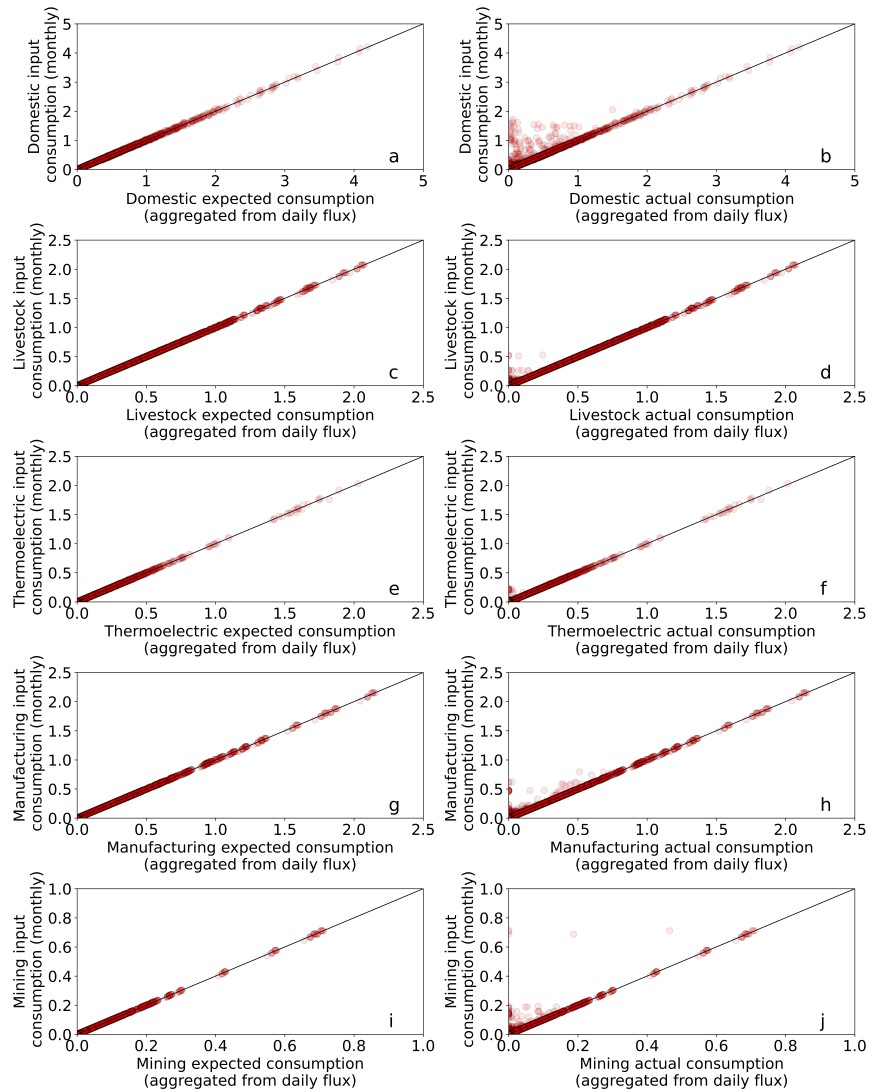

**Figure B4.** Comparison between expected monthly sectoral consumption values from input data, and monthly values aggregated from daily expected or actual consumption from model outputs. Each point represents the monthly value for a given grid cell. The points are plotted semi-transparently (alpha=0.5), therefore the more intense coloured parts simply indicate a larger concentration of values in that range. This plot was made by using the outputs for the year 2000 of the SectorWater experiment, and units for both axis are mm/month.

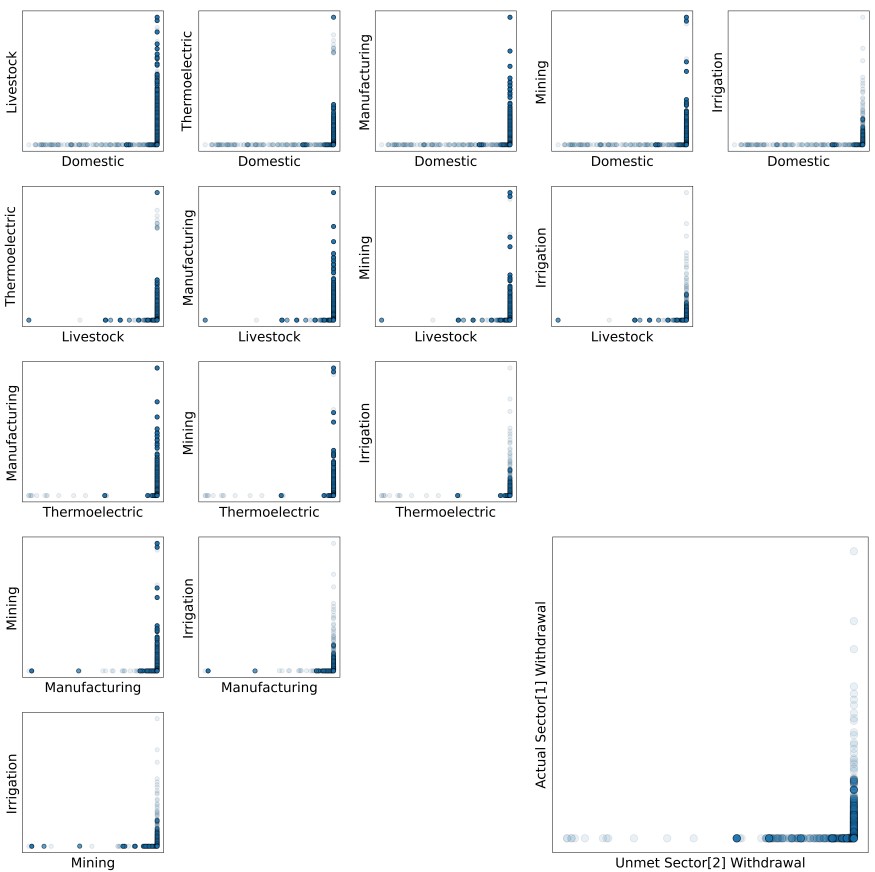

**Figure B5.** Evaluation of the sectoral competition algorithm, with each point representing a daily value at grid-cell level. The points are plotted semi-transparently (alpha=0.5), therefore the more intense coloured parts simply indicate a larger concentration of values in that range. The plot was made by sampling the first 30 days of the year 2000 from the SectorWater experiment. The intersection between the unsatisfied sectoral withdrawal of the sector higher in priority and the actual withdrawal of the sector lower in priority represents the 0 value.

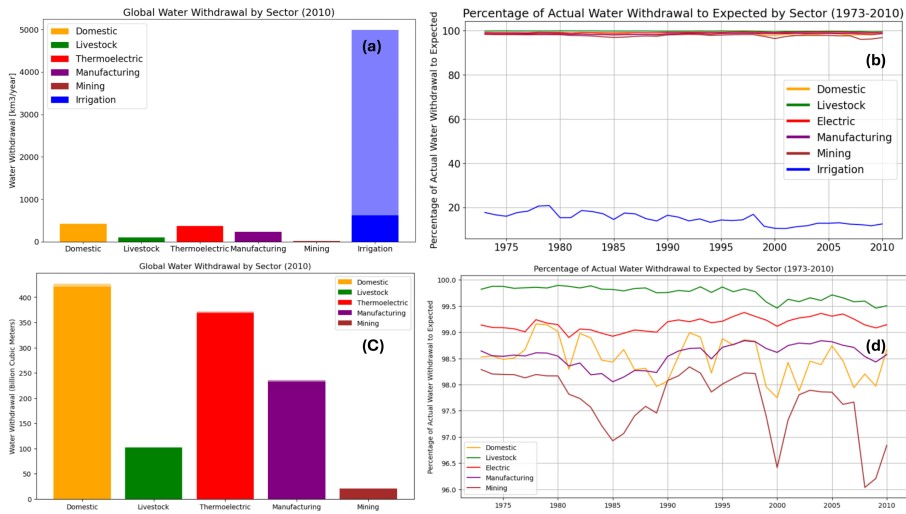

**Figure C1.** The figure shows annual global sectoral withdrawal (a, c) for the year 2010, and the time-series of global unmet sectoral demand throughout the period 1973–2010 (b, d). Non-irrigative sectors are separate from irrigation in the second row (c, d) for better visibility.

## Appendix C: Global Trends

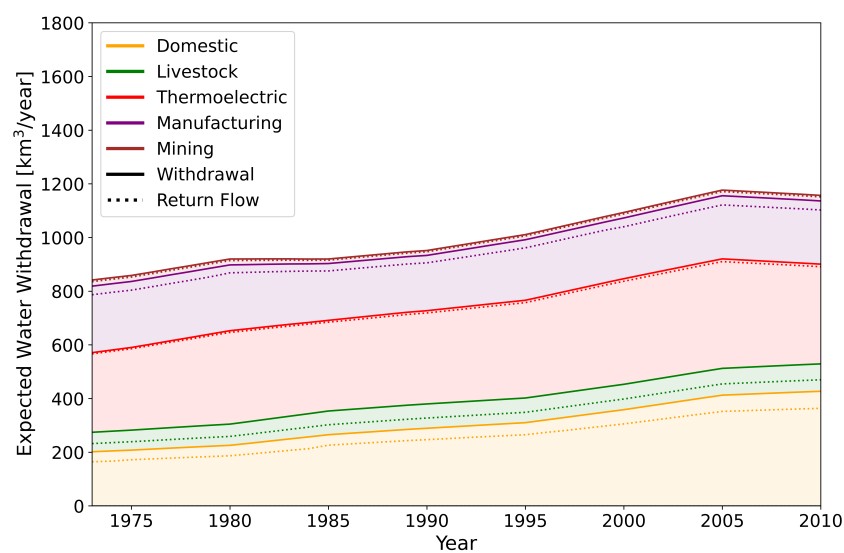

**Figure C2.** Annual global expected sectoral withdrawal and return flow (without irrigation) throughout the years 1973—2010. This figure was produced using the SectorWater experiment results. It should be noted that the consumption rate in the (Huang et al., 2018) dataset for the livestock sector is quite low when compared to other studies (0.4 vs 1.0). This explained by the usage of USGS estimated consumption rates globally, while other models simply assume 100% consumption.

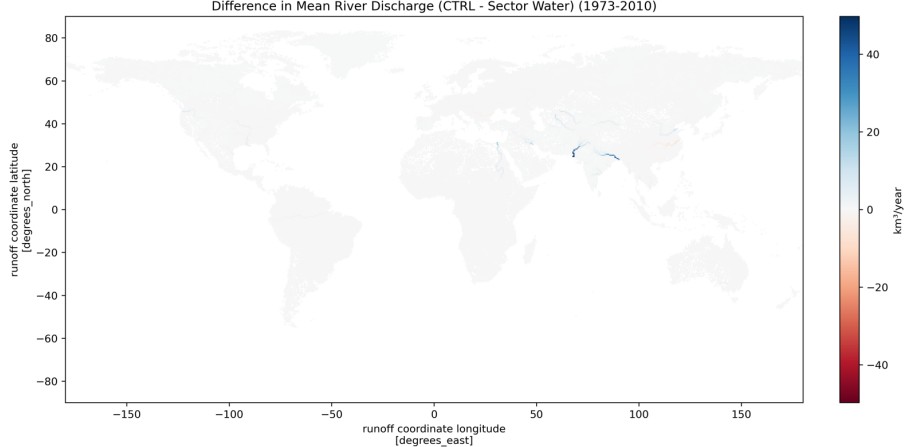

**Figure D1.** Difference in mean annual river discharge between CTRL and SectorWater experiments.

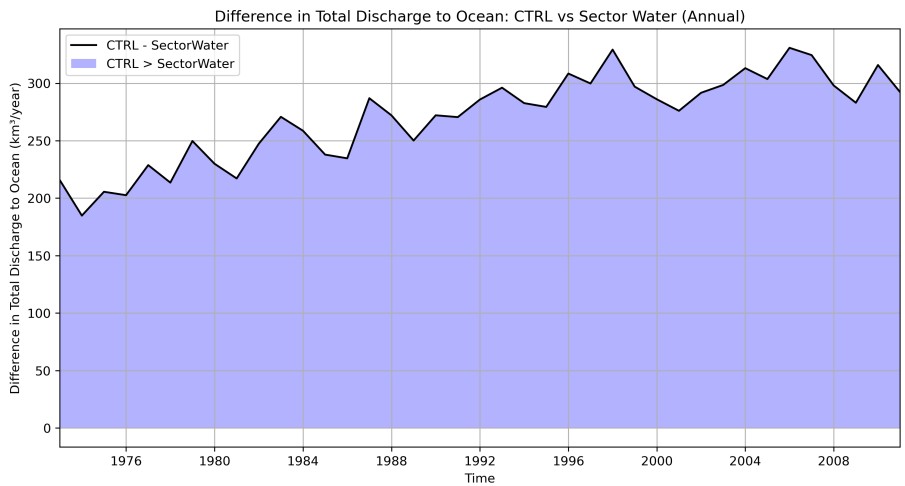

**Figure D2.** Difference in total annual river discharge to ocean for the period 1973–2010.

## Appendix D: Streamflow and ocean discharge changes

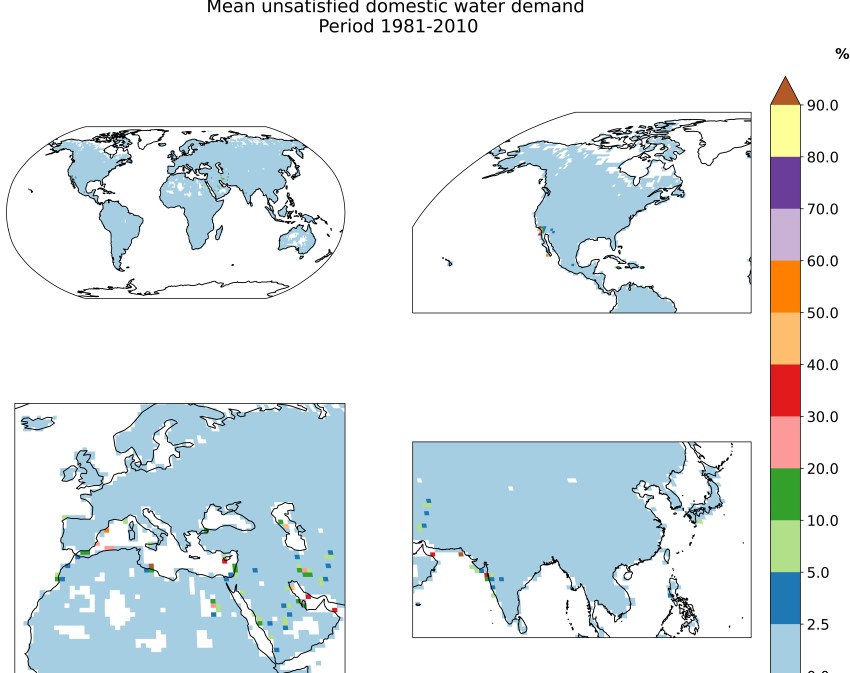

**Figure E1.** Fraction of unsatisfied domestic demand averaged over 30 years period (1981–2010). This figure was produced using the Sector-Water experiment results, by comparing expected vs actual withdrawal.

**Appendix E:  Water Scarcity**

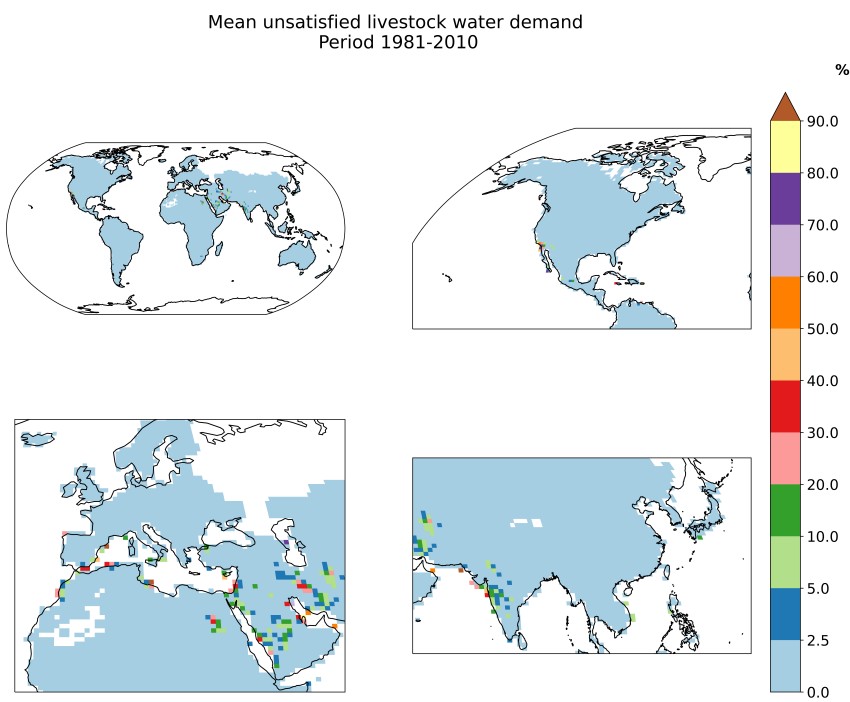

**Figure E2.** Fraction of unsatisfied livestock demand averaged over 30 years period (1981–2010). This figure was produced using the Sector-Water experiment results, by comparing expected vs actual withdrawal.

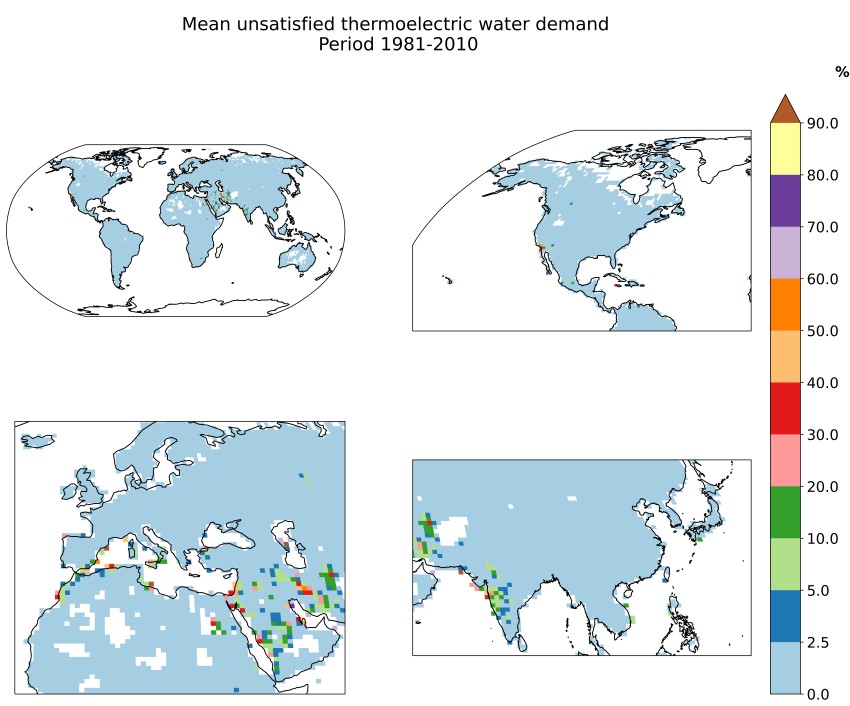

**Figure E3.** Fraction of unsatisfied thermoelectric demand averaged over 30 years period (1981–2010). This figure was produced using the SectorWater experiment results, by comparing expected vs actual withdrawal.

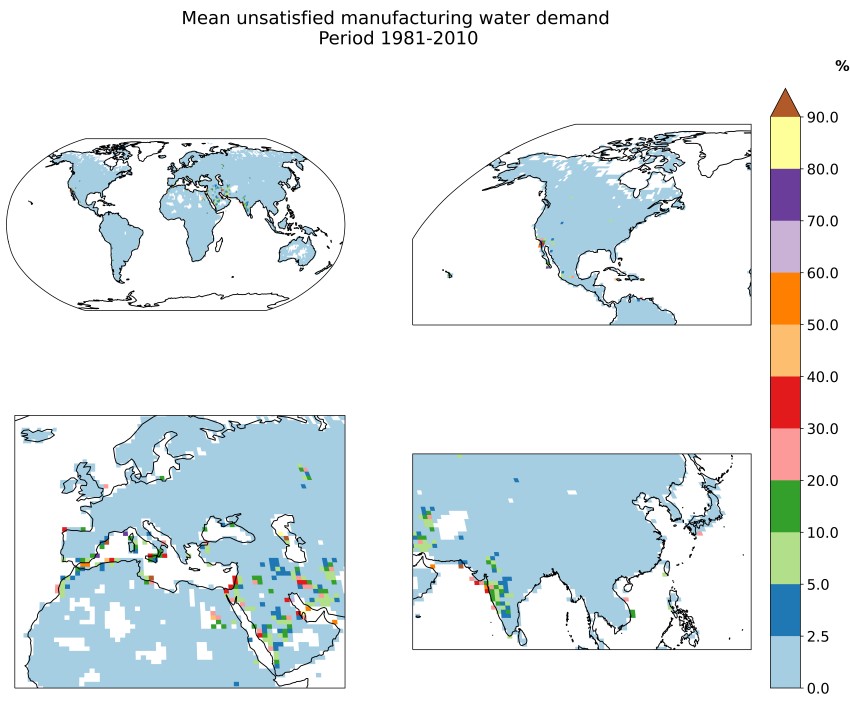

**Figure E4.** Fraction of unsatisfied manufacturing demand averaged over 30 years period (1981–2010). This figure was produced using the SectorWater experiment results, by comparing expected vs actual withdrawal.

Mean unsatisfied mining water demand
Period 1981-2010

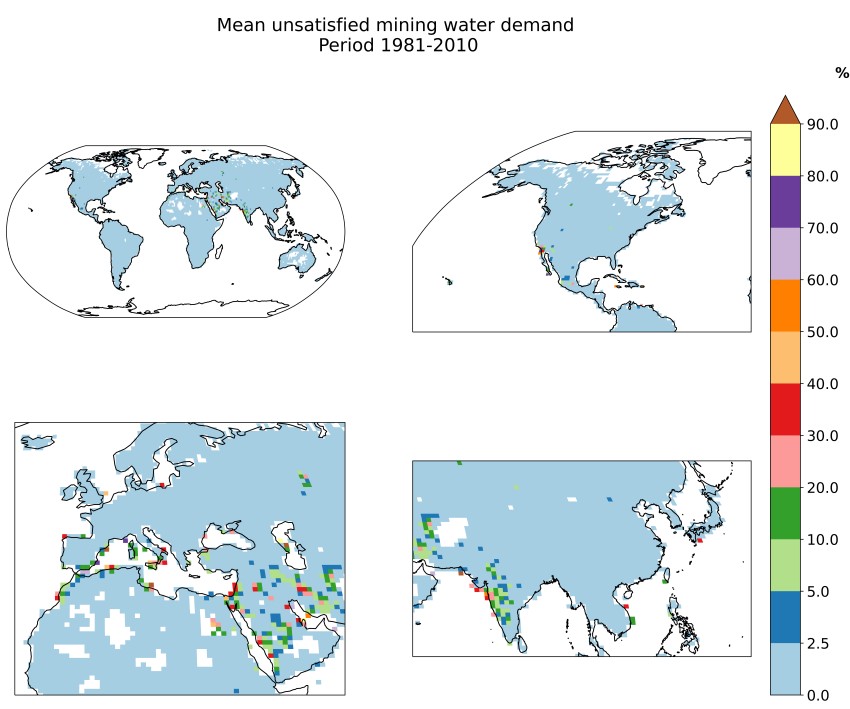

**Figure E5.** Fraction of unsatisfied mining demand averaged over 30 years period (1981–2010). This figure was produced using the Sector-Water experiment results, by comparing expected vs actual withdrawal.

Mean unsatisfied irrigation water demand
Period 1981-2010

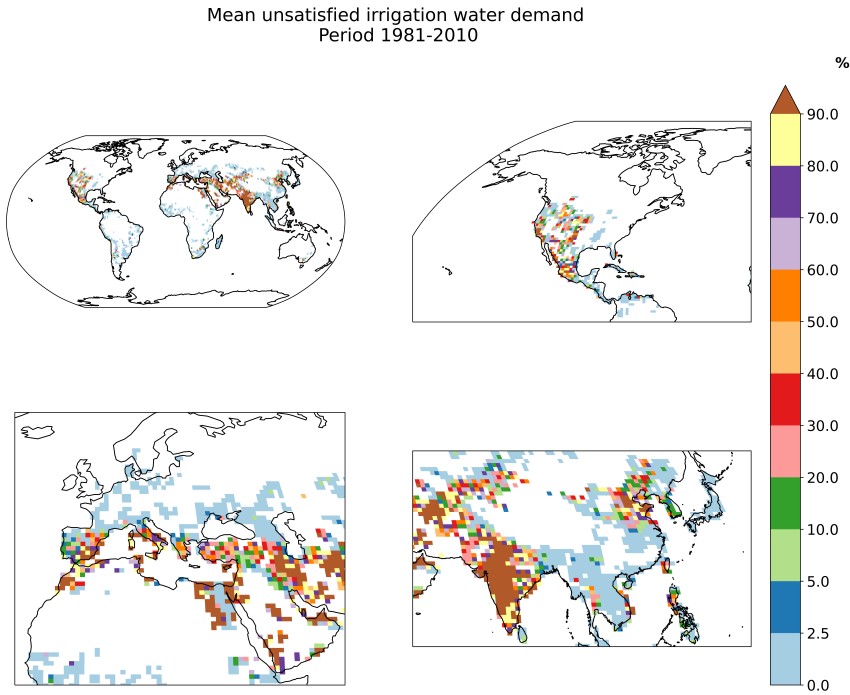

**Figure E6.** Fraction of unsatisfied irrigation demand averaged over 30 years period (1981–2010). This figure was produced using the SectorWater experiment results, by comparing expected vs actual withdrawal.

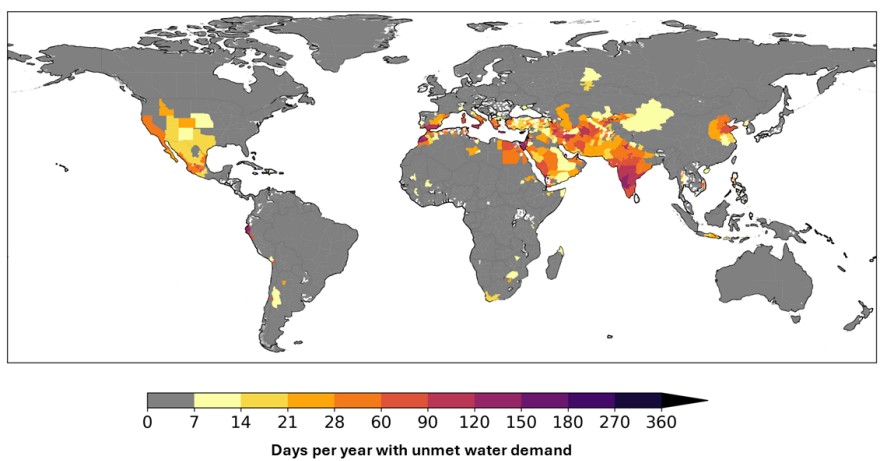

**Figure E7.** Average number of days per year, from 1973—2010, when modelled water supply was insufficient to meet the demand for all sectors. The values are aggregated at the first administrative divisions level within each country. Note the non-linear colour bar.

*Author contributions.* IST developed and implemented the new module, made the analysis and visualizations, and wrote the manuscript. EK, YY, SDH and IV provided technical support and feedback during model development stage. SR did additional testing of the module, and was responsible for the integration of the pull request into the main model branch. DL, YW, TT and WT supervised the work. All authors actively contributed to the project through regular meeting and provided critical feedback. All authors revised the final manuscript.

*Competing interests.* Some authors are members of the editorial board of journal Geoscientific Model Development.

*Acknowledgements.* The authors thank the National Center for Atmospheric Research, and in particular the Climate and Global Dynamics Laboratory, for hosting IST for a 1-month research stay. We would like to acknowledge high-performance computing support from Cheyenne (doi:10.5065/D6RX99HX) provided by NCAR's Computational and Information Systems Laboratory, sponsored by the National Science Foundation.

The authors thank the International Institute for Applied Systems Analysis, and in particular the Water Security group, for accommodating IST for a 3-months research stay and sharing their experience in developing global hydrological models.

The authors thank Sean Swenson, Naoki Mizukami, Andy Wood, Will Wieder, Ann Van Griensven and Steven Eisenreich for their critical feedback on the new module implementation, possible limitations and potential improvements.

ChatGPT (GPT-4, OpenAI's large-scale language-generation model) has been used to improve the writing style of this article. IST re-
viewed, edited, and revised the ChatGPT generated texts to his own liking and takes ultimate responsibility for the content of this publication.

This project has received funding from the European Union's Horizon 2021 research and innovation programme under the Marie Sklodowska-Curie grant agreement 956623, MSCA-ITN-ETN-European Training Network, inventWater (Inventive forecasting tools for adapting water quality management to a new climate) Project.

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
