# Peer review of "Bridging the gap: a new module for human water use in the Community Earth System Model version 2.2.1"

_EGUsphere, 2024_

## Author Comment (AC2)

**Bridging the gap: a new module for human water use in the Community Earth System Model version 2.2.1**

Geoscientific Model Development

July 3, 2024

S. I. Taranu[1], D. M. Lawrence[2], Y. Wada[3,4], T. Tang[3,4] E. Kluzek[2], S. Rabin [2] Y. Yao[1], S. J. De Hertog[1,5], I. Vanderkelen[6,7,8], and W. Thiery[1]
sabin.taranu@vub.be

[1] Vrije Universiteit Brussel, Department of Water and Climate, Brussels, Belgium
[2] National Center for Atmospheric Research, Climate and Global Dynamics Laboratory, Boulder, CO, USA
[3] Biological and Environmental Science and Engineering Division, King Abdullah University of Science and Technology, Thuwal, Saudi Arabia
[4] International Institute for Applied Systems Analysis, Laxenburg, Austria
[5] Universiteit Gent, Department of Environment, Q-ForestLab, Ghent, Belgium
[6] Wyss Academy for Nature at the University of Bern, Bern, Switzerland
[7] Climate and Environmental Physics division, University of Bern, Bern, Switzerland
[8] Oeschger Centre for Climate Change Research, University of Bern, Bern, Switzerland

**Contents**

**Abstract**

This response letter contains numbered figures and references to these figures. To prevent confusion, the figures embedded within this response letter are called illustrations. Finally, the following convention is applied to denote modification in the original manuscript: new text.

**1 Reviewer 1**

We thank the Reviewer 1 for the constructive feedback to improve the manuscript. Below, we address every comment carefully and explain the corresponding changes in the manuscript.

**1.1 Major comments**

> **Reviewer 1 Comment 1**
>
> I think the manuscript would be improved by focusing on just the emergent behaviors of the new water sector fluxes, and removing peripheral or redundant information, especially in regards to irrigation, which has been discussed in previous studies. Specifically, I would like to see more details of how the prioritization scheme cascades through the different sectors, especially in regards to water scarcity. For instance, figure 11 shows the combined days per year of unmet demand, and it would be informative to see similar plots for each sector, highlighting where the occur and the relative magnitudes of each sector. This would be similar to appendix D, but I think showing days per year may be more relatable to the reader than the fractions of unmet demand.

**Response** We thank the reviewer for these suggestions. Concerning the irrigation results, we will significantly reduce mentioning them across the text and focus more on the new model capabilities. We also agree that having some more analysis exploring the cascading effect from the sectoral competition could be of interest. We expanded the Section 3.5 on Water Scarcity with the following figures:

[Figure]

Illustration 1: Average number of days per year, from 1973–2010, when modelled water supply was insufficient to meet the demand for individual sectors. Note the non-linear colour bar.

Illustration 1 (and 2 for Supplementary materials), are done exactly as the reviewer sug-

[Figure]

Illustration 2: Average number of days per year, from 1973–2010, when modelled water supply was insufficient to meet the demand for individual sectors. The values are aggregated at the first administrative divisions level within each country. Note the non-linear colour bar.

gested. But we also further expanded our analysis by showcasing how the model could explore trends in water scarcity at the country level with Illustrations 3-5. To accommodate these results, we updated the text for the corresponding section 3.5 on Water Scarcity in the reviewed version of the manuscript.

[Figure]

Illustration 3: Fraction of the country with at least one day per year when modelled water supply was insufficient to meet the demand for individual sectors. The results are provided as a 10 years rolling average for better representation of long-term trends.

[Figure]

Illustration 4: The average number of days per year, countries are exposed to unsatisfied sectoral demand. The average is calculated only using the gridcells which experience water scarcity conditions. The results are provided as a 10 years rolling average for better representation of long-term trends.

[Figure]

Illustration 5: Countries sectoral withdrawal for the years 1973–2010. The results are pro-vided as a 10 years rolling average for better representation of long-term trends.

> **Reviewer 1 Comment 2**
>
> I encourage the authors to remove most of section 3.1 and figures 3-5. Confirmation of conservation is obviously necessary during model development, but a single sentence could indicate that this was successfully performed.

**Response** We agree with the reviewer, and moved the figures 3-4 to the supplementary materials. For the figure 5, we would prefer to keep it if possible, for reasons we will provide in Reviewer 1 Comment 5. We also significantly shortened the section 3.1, while maintaining track of what was done for the validation process.

> **Reviewer 1 Comment 3**
>
> In CESM mass/energy conservation violations cause the model to abort, therefore a completed simulation is proof enough.

**Response** The reviewer is right and we now mention this in the text instead of the previous explanation:

> Finally, CLM5 ensures mass and energy conservation by aborting the simulation when this is violated.

> **Reviewer 1 Comment 4**
>
> That being said, I find the authors comment that the errors in regridding the input data were 1-2% ( line 245) to be worrisome. Conservative regridding should be much more accurate than that.

**Response** We understand how 1-2% may seem large, taking into account that we do a conservative remapping. We think that these differences are attributed to the upscaling effect from passing to a lower resolution. For example, the coastline definition may change substantially, while some insular countries may disappear when comparing the 0.5x0.5 and 0.9x1.25 land mask. The procedure we used for remapping was the same as for the other inputs in CLM5. Since the remapping is done on a binary land mask (land or water) instead of a fractional one, it is not possible to conserve the global withdrawal amount, because of changes in land area.

To reflect this point, we added to the text:

> In general, the remapping procedure is found to be conservative (Fig. B1), with relative errors of about 1-2% explained by upscaling effects when passing to a lower resolution land mask (Fig. B2).

To conserve the global withdrawal amount, we could redistribute the error. For this, we would take the weights after remapping (ratio between gridcell to global withdrawal) and use it to downscale the correct global amounts for individual sectors.

We want to highlight that this difference is related to the pre-processing of input data. The module itself introduces no error. In the context of this study, we think that the 1-2% difference in global withdrawal, remains tolerable and would not change our results. But we will consider this issue, and apply the proposed correction for our future applications.

> **Reviewer 1 Comment 5**
>
> Similarly, I thought figure 4 could be removed and a simple statement indicating that the prioritization algorithm worked properly would provide equivalent information. Given that CLM5 and MOSART have previously been coupled in a conservative manner, one would expect that any flux passed between the two would be treated the same way, and thus figure 5 is unnecessary.

**Response** We agree and moved the figure 4 to the supplementary materials, while shortening the phrase mentioning this result.

At the same time, if possible, we would like to keep the figure 5 displaying the outputs from CLM5 and MOSART, as well as the associated explanation of the successful coupling. We have 2 reasons for this.

First, the coupling support for the non-irrigative sectors didn't simply involve adding new fluxes for the coupler, but also writing a dedicated module for this (map_lnd2rof_sectorwater_mod.F90 module on GitHub). We now reflect this better in the text (Sect 2.2):

> Conversely, to send the information concerning each sector withdrawal and return flow to the MOSART model, a new module was implemented in the coupler codebase, that divides the sectoral fluxes across all main channels within the corresponding CLM5 grid cell in accordance to their relative weight in current water storage capacity... Same approach was originally implemented for the irrigation without the return flow part. The new module being therefore a generalization for the remaining sectors.

Secondly, in our experience sharing this development with other researchers from Land Surface and Global Hydrological modelling communities, the questions of how the withdrawal, consumption, and return flows are treated across the land-routing models were very common. The figure 5 in particular often invited questions, because it is quite common for the routing model to operate at a higher resolution than the associated land model, so the information about how the fluxes are remapped from one model to the other is of relevance.

> **Reviewer 1 Comment 6**
>
> Discussions of the climate impact of irrigation have previously been published for CLM5, so I would remove this type of discussion, and instead focus on how irrigation is changed by the addition of the other sectors. For instance, figure 9 is largely a recapitulation of earlier results; I would prefer to know if there are any locations where the other water sectors have a significant impact. In that regard, the authors might wish to include irrigation in the CTRL simulation.

**Response** We agree with the reviewer that the section on climate impacts is too focused on the irrigation results, while not explaining well why there is no significant effect from the non-irrigative consumption. To improve on this, we added:

There are a few reasons why we don't find a significant effect for the non-irrigative sectoral consumption on the selected surface variables in our experiment. For example, for non-irrigative sectors: the cumulative consumption is quite small; it is distributed over the year and the entire day (24h), and independently of soil moisture conditions; distributed over a larger area (natural vegetation column). As a consequence, the impact of non-irrigative sectors on the climate is insignificant. For irrigation: the water is applied for parts of the year when the crops grow, and only when there is a moisture deficit; the irrigation window is short, applied over 4 hours at 6am local time; locally applied over the columns with irrigated crops; the total irrigation amounts being very large (e.g. maximum cumulative grid cell consumption for non-irrigative sectors is only at the low end of irrigation grid cell consumption in Fig. 9). These conditions ensure that irrigation will have a significant impact on the local climate, since a significant amount of water is provided locally over a short period of time in a moment when evaporation and plant transpiration is limited by water availability.

These findings underscore irrigation's principal role as a climate forcing in relation to human water use (McDermid et al., 2023), with other sectors contribution being negligible at large scale. While not for their climate impact, incorporating other sectors' abstractions into Earth System Models (ESMs) as suggested by Nazemi and Wheater (2015), remains important for evaluating water scarcity in present and future climates, as well as the changes of surface and groundwater storage resulting from sectoral withdrawal and consumption.

Concerning adding new simulations to the paper, to better isolate the effect of other sectors, we think that this might not be necessary. Before deciding on the experiments, we considered several options.

For example, we did run a shorter run with the same CTRL, but with the new sectors only, without irrigation to try to isolate the impact of other sectors. But we couldn't find a significant effect on the climate, and this is why we finally went for this setup, where we wanted to show the contrast between the role of irrigation and other sectors on modulating local climate. This wouldn't be possible if we add irrigation to the control. Our main result here is to highlight that implementing other sectors in Earth System Models (ESMs) might not provide any added value in terms of climate modulation (at least at large scale). To our knowledge, this result was never presented before.

**1.2   Minor comments**

> **Reviewer 1 Comment 7**
>
> Line 1: natural climate variability should be mentioned in addition to shifting climate patterns, e.g. in places like east Africa, where decadal variability has significant impacts historically.

**Response** We agree that natural climate variability is an important trigger of water scarcity in certain regions. At the same time, we realized that what we really wanted to say is not what causes water scarcity, but rather what is exacerbating the existing problem. So we reformulated the text:

> Water scarcity is exacerbated by climate change, as well as rising water usage, yet state-of-the-art Earth system models typically do not represent human water demand.

At the same time, we now mention climate variability as a source for water scarcity in Sect. 3.5 during the interpretation of the new figures.
* * *
**Reviewer 1 Comment 8**

Line 10: what is meant by "captures global patterns"?
* * *
**Response** We judged that this phrase does not provide any added value, so we ended up taking it out from the abstract.
* * *
**Reviewer 1 Comment 9**

Line 48: 'spatial' resolution
* * *
**Response** Corrected text:

> At the finer end, some models have spatial resolutions as detailed as a few kilometres, making short-term weather predictions and analysing specific hydrological processes possible (Prein et al., 2015).
* * *
**Reviewer 1 Comment 10**

Line 57: provide a reference to 'demonstrating higher skill' in CMIP
* * *
**Response** We added the reference, which is a figure from the IPCC AR6 Science Basis Technical Summary. The figure compares model resolution, complexity, and correlation with observations for CMIP3, CMIP5, CMIP6 and HighRes ensembles. It shows a systematic increase in correlation for each new CMIP generations for a range of climate variables. Here is the adjusted text:

> The field of Earth system modelling has seen important progress between different iterations of the Coupled Model Intercomparison Project phases, demonstrating higher skill in matching observations across relevant climate change indicators (IPCC AR6 Technical Summary: The Physical Science Basis, 2021, Fig. TS.2). This can be attributed to running the models at higher resolutions, increasing model complexity, and improving the representation of physical, chemical, and biological processes (IPCC AR6 Technical Summary: The Physical Science Basis, 2021).
* * *
**Reviewer 1 Comment 11**

Line 89: perhaps state that coupler allows components to have grids with different spatial resolutions

**Response** This is a very useful suggestion, since even in our simulations, CLM5 and MOSART have different spatial resolutions (also time-step). Here is the adjusted text:

> These individual models, which may operate at different spatial resolution and time step, interact and exchange states and fluxes via a coupler (Danabasoglu et al., 2020).
* * *
**Reviewer 1 Comment 12**

Line 104: what happens if demand is not met?
* * *
**Response** There are 2 options. If the user selected in the namelist, limit_irrigation_if_rof_enabled = True, then the irrigation amount will be limited to 90% of current river water storage. Else, the irrigation amount might exceed the river storage. In this case, river storage goes to zero till updated at next timestep, and the additional missing amount is compensated by ocean water.

The first setup is more realistic, and the one we used in our SectorWater experiment. This is how we can track unmet demands for irrigation. But the second setup was also extensively used in research, primarily in studies where the effect of irrigation on the climate is studied. We mention this in Sect. 3.2 (lines 281 till 295).

We agree that it will be relevant to add this information here, so we updated the text as follows:

> Once the irrigation water demand is met by abstraction from the river network, it is applied over the surface soil across irrigated crop columns (Fig. A1). This arrangement allows the water to contribute to crop growth and become a part of the surface water-energy balance through processes like evapotranspiration, runoff, and infiltration (Fig. A2). In case there is not enough water to fully satisfy irrigation requirements, the model have 2 options. First, to limit irrigation abstraction to 90% of current river storage, which helps maintain some environmental flow requirements. Or secondly, abstract as much as needed, with the missing part being compensated by ocean water. While unrealistic, the second setup was successfully used in studies where having total irrigation requirements satisfied is more relevant (Thiery et al., 2017; Yao et al., 2022).
* * *
**Reviewer 1 Comment 13**

Line 132: MOSART is not aware of the CLM grid; states and fluxes are sent to the coupler. Line 135: again, models are not aware of each others' grid structures in CESM; overlapping areas in each grid are calculated in the coupler, so this should be re-worded to accurately reflect how the coupling is done.
* * *
**Response** Thanks for noticing this inaccuracy. Indeed, this is done by the coupler. Here is the corrected text:

> The information about how much water is potentially available for sectoral use is provided by the coupler to the CLM5 model at grid cell level by calculating the corresponding total river network storage in the MOSART model. Conversely, to send the information concerning each sector withdrawal and return flow to the MOSART model, a new module was implemented in the coupler codebase, that divides the sectoral fluxes across all main channels within the corresponding CLM5 grid cell in accordance to their relative weight in current water storage capacity...

**Reviewer 1 Comment 14**

Line 141: 'idealized' may be more appropriate than 'real-world' given the simple prioritization scheme; could just say 'to diagnose instances of water scarcity'.

**Response** We updated the text as suggested:

To simulate idealized scenarios to diagnose instances of water scarcity, the new module implements a basic sectoral priority algorithm for situations when water availability is inadequate to meet the full sectoral demand.

**Reviewer 1 Comment 15**

Line 143: are the Roman numerals used elsewhere? If not, why are they used?

**Response** We opted to remove the numerals as suggested.

**Reviewer 1 Comment 16**

Line 158: a discussion of the dominant modes of water use for each sector would be useful. Water used for cooling for thermoelectric and water for mining purposes might be expected to be returned, but why is return flow so large for livestock? Does figure C1 indicate that 1/2 of livestock water is return flow?

**Response** We now added to the section 2.3 (Input data) the following paragraph:

Domestic water withdrawal encompasses the use of water for various indoor household activities, including drinking, food preparation, bathing, laundry, dishwashing, and toilet flushing. It also includes outdoor uses like watering lawns and gardens, as well as water use by public sector and service industry. Electricity water withdrawal refers to the water used for cooling thermoelectric and nuclear power plants. Water withdrawal for mining is utilized in the extraction of minerals, which can be solids (like coal), liquids, or gases (such as natural gas). In manufacturing, water withdrawal serves multiple purposes including fabricating, processing, washing, cooling, or transporting products, incorporating water into products, and meeting the sanitation needs within the manufacturing facilities. These sectoral water withdrawal categories are consistent with the work of (Liu et al., 2016), and described in (Huang et al., 2018).

Concerning the livestock sector, the consumption rate is fixed globally to 0.4. This may seem indeed very low, as many other models assume livestock sector to be fully consumptive. In their paper, (Huang et al., 2018) explained that in general the consumption rates estimates

are rarely provided by national/sub-national statistics, except of US which provided such estimates through USGS. So they ended up using the US based estimates globally.

To capture this detail, we added to the figure C1 caption:

> Annual global expected sectoral withdrawal and return flow (without irrigation) throughout the years 1973—2010. This figure was produced using the SectorWater experiment results. It should be noted that the consumption rate in the (Huang et al., 2018) dataset for the livestock sector is quite low when compared to other studies (0.4 vs 1.0). This is explained by the usage of USGS estimated consumption rates globally, while other models simply assume 100% consumption.

**Reviewer 1 Comment 17**

Line 212: the discussion of spatial and temporal patterns doesn't seem appropriate here, given that this is a hypothetical case for a single location. Figure 2: I would find it more natural to indicate the events in temporal order, since the prioritization works vertically in this case, i.e. a,d,e,b,c.

**Response** We understand the reviewer position here. The main reason we present the results in this order and with the addition of details regarding potential spatial variability in represented sectors, is because our main objective was not only to show the prioritization algorithm, but also provide a larger overview of how the model works and what situations may occur.

**Reviewer 1 Comment 18**

Line 229: as previously indicated, I would include irrigation in CTRL, as irrigation is currently modeled by CLM5. Also, it should perhaps be mentioned that sectors that have large return flow to runoff would not affect climate.

**Response**

We would like to refer the reviewer to the Reviewer 1 Comment 6 as we address the reasons behind the current experiment protocol in details there.

Concerning the return flow and consumption for the non-irrigative sectors, we keep the return flow fluxes separate for each sector, but the consumption for all the new sectors is summed up into a single flux. We decided to keep the return fluxes separate because later on, this could be used to estimate various pollutants loads if a water quality module would be added later to the model (we mention this in the Sect. 4 on limitations). For the consumption part, since we didn't see any significant climate modulating effect or other future benefits, we decided to sum all contributions in one flux to reduce computational and memory costs. This means we cannot really differentiate the potential consumption impact of individual sectors, except if we run experiments with one sector at a time.

> Reviewer 1 Comment 19
>
> Line 266: Section 3.2 begins by discussing trends, but mainly discusses irrigation and river-only vs unlimited simulations. The first paragraph could be improved by discussing the statistics in more detail rather than trying to compress all the information into 1 or 2 sentences.

**Response** We extended the paragraph with some more detailed information:

> Over the period 1973–2010 (Fig. 6), sectoral water withdrawals shows mostly a steady increase, followed by a slight decline towards 2010. While irrigation is consistently the largest contributor, other sectors account for a substantial share, especially looking at the actual withdrawal. By 2010, the cumulated non-irrigative expected sectoral withdrawal and consumption reached, 1157 and 171 km3/year respectively, which represent an increase of 315 km3/year or 36% for the period 1971–2010. The total global expected and actual water withdrawal increased by 110% and 43% respectively. Over the same period, in the SectorWater simulation, the expected (actual) irrigation withdrawal accounts for 79% (36%) of total water withdrawal, followed by domestic with 6% (19%), thermoelectric with 8% (23%), manufacturing with 5% (14%) and finally livestock and mining together at about 2% (8%).

We also added another figure to the supplementary materials (Illustration 6). We refer to this figure a little later in another paragraph:

> When analysed globally based on the SectorWater experiment results, CLM5 supplies only 10-20% of what is globally requested (Illustration 6.d), compared to >96% for all the other sectors.

[Figure]

Illustration 6: The figure shows annual global sectoral withdrawal (a, c) for the year 2010, and the time-series of global unmet sectoral demand throughout the period 1973–2010 (b, d). We separated the other sectors from irrigation in (c, d) for better visibility.

> **Reviewer 1 Comment 20**
>
> Line 313: the word 'prioritize' may need context; doesn't everyone prioritize domestic water use? Would industrial water use ever be prioritized over domestic water use? (I am thinking of domestic water for consumption and sanitation).

**Response** Here is the improved text:

> Different socio-economic and climatic conditions lead to diverse water use profiles (see the map in Fig. 8). Agriculture dominated water use is prevalent in many African, Asian, and South American countries, whereas industrialized nations in Europe and North America show a greater emphasis on industrial water use. Arid regions like the Middle East have higher domestic and agricultural water use due to increasing population and urbanization, while the high evapotranspiration rates and limited soil moisture makes irrigation essential for crop growth (World Bank, 2017).

> **Reviewer 1 Comment 21**
>
> Line 317: the wording is vague: does 'trends shifted' mean the location of trends changed, or that trends changed in time?

**Response** Here is the improved text:

> From 1973 to 2010, water withdrawal trends changed, reflecting socio-economic progression, population dynamics, and changes in regional climate.

We then proceed to explain how the trends changed in the text.

> **Reviewer 1 Comment 22**
>
> Line 331: is there any reason to think that irrigation would act differently in this simulation? I would rather see the irrigation included in the CTRL simulation and a discussion of the interactions between irrigation and the other sectors. Otherwise, this is not novel and has been reported previously.

**Response** In terms of climate impact, the effect of irrigation wouldn't be different in the sense of direction, only magnitude, which we didn't focus on because outside the scope of the study. What we showed is for example that global actual irrigation withdrawal is lower in our experiment (about 400 km3/year) than when only irrigation is activated (about 910km3/year). We did this in Sect. 3.2 on "Historical trends in global sectoral withdrawal", by comparing our results to the Yao et al. (2022) study. Partially, this is caused by the sectoral competition, and partially simply because we limit sectoral supply to 90% of river water availability.

Concerning the model setup and the novelty aspect of our result, we would like to refer the reviewer to the Reviewer 1 Comment 6 as we address these issues in details there.

> **Reviewer 1 Comment 23**
>
> Figure 10: I found it hard to compare the two maps due to the scale. Perhaps showing the ratio would be more insightful?

**Response** We considered this option, but the main issue is that there are many grid cells where the irrigation withdrawal/consumption is 0, so presenting the results as ratio between the two will be problematic. Similarly, grid cells with very small values, may be misrepresented with very high ratios.

The main point we tried to show was that the cumulative sectoral consumption map doesn't correlate well with climate impacts (Fig 9). Also, irrigation consumption is often much larger. We now mention this in the figure caption:

> Spatial distribution of mean annual consumption for all sectors (excluding irrigation) alongside annual irrigation withdrawal, both used as predictors in the regression depicted in Fig. 9. It can be noted that irrigation withdrawal correlates much better with the surface climate variables in Fig. 9. While the irrigation withdrawal is in general much larger than non-irrigative sectoral consumption, this is especially the case for the points where a climate modulating effect is observed.

> **Reviewer 1 Comment 24**
>
> Line 345: in section 3.5, be clear about what statements are model results vs observed conditions. Also, explain how individual sectors are impacted at the locations mentioned for instance, is the statement that tourists increase demand reflected in a domestic shortfall or other sectors?

**Response** We updated the section 3.5 in line with the reviewer comment.

> **Reviewer 1 Comment 25**
>
> Line 366: some details from the referenced studies would be informative here.

**Response** We updated the section 3.5 in line with the reviewer comment.

> **Reviewer 1 Comment 26**
>
> Line 382: is it true that confined aquifers are not supported? There is a variable in CLM called "qflx_gw_con_irrig_col" described as "confined groundwater irrigation flux".

**Response** While the variable was added and the code necessary to allow confined groundwater abstractions is present, the confined groundwater resource itself is not represented. In this case, the code was simply developed (not by the authors of this publication) at the same time as the support for unconfined groundwater abstractions, in anticipation that this feature might appear in the future.

To add support for confined groundwater abstractions, from current state of the model, it will be enough to define confined groundwater storage and its dynamics (sources/sinks).

> **Reviewer 1 Comment 27**
>
> Line 399: this approach has been attempted previously: Anderson et al., 2015, Using satellite-based estimates of evapotranspiration and groundwater changes to determine anthropogenic water fluxes in land surface models, GMD, DOI:10.5194/gmdd-8-3565-2015

**Response** We read the recommendation, and it is an excellent match for what we tried to convey. We now added the reference to the text.

> **Reviewer 1 Comment 28**
>
> Line 484: what is meant by 'adeptly captures'; isn't the water use an input to the model? The interesting thing is not necessarily which sector dominates based on magnitude, but how each is affected by scarcity.

**Response** We updated the section accordingly:

> While irrigation is the dominant sector for water use, we showed that for many regions other sectors, like domestic or industrial, dominate. This challenges the usual focus on irrigation in Earth System Models and points to the increasing importance of non-agricultural water demands in areas experiencing rapid population growth and socio-economic development. Through the implementation of all major water use sectors, it is now possible to study water scarcity in more details, by analysing sector specific unmet demands and impacts.

> **Reviewer 1 Comment 29**
>
> Line 497: 'uniquely' is unlikely to continue, perhaps simply say 'is well positioned'

**Response** We agree and changed the text as suggested:

> In this regard, with the new sectoral water capability, the CLM5 model is well positioned for research on water use and scarcity, paralleling the capabilities of Global Hydrological Models (GHMs).

**2 Reviewer 2**

We thank the Reviewer 2 for the constructive feedback to improve the manuscript. Below, we address every comment carefully and explain the corresponding changes in the manuscript.

**2.1 Major comments**

> **Reviewer 2 Comment 1**
>
> Perturbations on water and energy balance due to the new sectoral water consumption model:
>
> While significant variations are expected due to the new sectoral module, no evidence for water and energy conservations is listed in the paper. The authors need to show evidences by comparing the water/energy balance between the CLM version with and without the sectoral water module and are also encouraged to add some more explanation regarding the water and energy coupled balance induced by a sectoral water. For example, given that this is a global-scale study, the authors can show how the streamflow at the coastal area is varied after the implementation of the new module.

**Response** In the CESM/CLM models, the simulation is aborted if mass/energy conservation is violated, so for this reason we did not focused too much on this aspect. We now added this clarification to the text:

> Finally, CLM5 ensures mass and energy conservation by aborting the simulation when this is violated.

We also agree with the reviewer that having some analysis dedicated to streamflow changes would be interesting. In line with what was suggested, we added two figures to the supplementary materials. Illustration 7 shows, for the SectorWater experiment, a significant decrease in the mean annual streamflow for most large rivers due to the cumulated consumption across the river basins. Illustration 8, shows the changes between CTRL and SectorWater experiments in terms of total annual river discharge to ocean. The cumulated sectoral consumption (irrigation included) is responsible for a decrease of river discharge to oceans of about 200-300 km3/year.

We now mention this in the text:

> While not for their climate impact, incorporating other sectors' abstractions into Earth System Models (ESMs) as suggested by Nazemi and Wheater (2015), remains important for evaluating water scarcity in present and future climates, as well as the changes of surface and groundwater storage resulting from sectoral withdrawal and consumption. For example, comparing the SectorWater and CTRL experiments, shows a significant decrease in mean annual streamflows for most major rivers (Illustration 7), as well as a decrease in total annual river discharge to ocean of about 300 km3/year by the year 2010 (Illustration 8).

[Figure]

Illustration 7: Difference in mean annual river discharge between CTRL and SectorWater experiments.

[Figure]

Illustration 8: Difference in total annual river discharge to ocean for the period 1973–2010.

> **Reviewer 2 Comment 2**
>
> Implication:
>
> Given the almost zero correlation between energy balance variations and water consumption (green dots in figure 9), how the reader should understand the importance of the results? The authors just briefly talked about the role of irrigation (which is not a focus of this paper) but none exists for explaining the relationship between the 'Sectoral cons' and energy balance variations. More elaborated and process-based explanation should be added in the revised manuscript to get reader more ideas about the implication of this work. Is their approach converting the consumed water into evaporation still effective with this results?

**Response** We agree with the reviewer and now added the following process based explanation to the corresponding section:

> There are a few reasons why we don't find a significant effect for the non-irrigative sectoral consumption on the selected surface variables in our experiment. For example, in the case of the irrigation sector in CLM5 the following things apply: the water is applied for parts of the year when the crops grow, and only when there is a moisture deficit; the irrigation window is short, applied over 4 hours at 6am local time; locally applied over the columns with irrigated crops; the total irrigation amounts are very large (e.g. maximum cumulative grid cell consumption for non-irrigative sectors is only at the low end of irrigation grid cell consumption in Fig. 7). These conditions ensure that irrigation will have a significant impact on the local climate, since a significant amount of water is provided locally over a short period of time in a moment when evaporation and plant transpiration is limited by water availability. At the same time, the situation is very different for non-irrigative sectors: the cumulative consumption is quite small; it is distributed over the year and the entire day (24h), and independently of soil moisture conditions; distributed over a larger area (natural vegetation column). As a consequence, the impact of non-irrigative sectors on the climate is insignificant.

We also briefly discuss the implications of our findings. Specifically, we suggest that sectors beyond irrigation should still be included in Earth System Models (ESMs), but not for their effects on climate modulation. To our knowledge, this is a new result which was not shown before.

> These findings underscore irrigation's principal role as a climate forcing in relation to human water use (McDermid et al., 2023), with other sectors contribution being negligible at large scale. While not for their climate impact, incorporating other sectors' abstractions into Earth System Models (ESMs) as suggested by Nazemi and Wheater (2015), remains important for evaluating water scarcity in present and future climates, as well as the changes of surface and groundwater storage resulting from sectoral withdrawal and consumption.

> **Reviewer 2 Comment 3**
>
> Values:
>
> While the values for annual actual withdrawal is presented, I wonder if the authors are really certain about the values. The values ranges up to 700 mm/year. Even for some wet area with annual precipitation around 1,500 mm/y – 2,000 mm/y, if we apply relatively higher runoff ratio such as 0.7, the streamflow could ranges up to 1400 mm/y. I would think this is an extreme case having abundant water resources and even for this case the 700 mm/year consumption of water is the half of the total streamflow. Furthermore, if we sum the (a), (b), (c), ... (f), what is the total consumption of water at the grid-scale? Can you provide the spatial distribution of the total sum of withdrawal?

**Response** While estimates of water withdrawal are uncertain and can heavily depend on human behaviours and modelling approaches, our estimates are consistent with the source of the data we used. For example, for the non-irrigative sectors a confirmation check can be made with Fig. 3 from the Huang et al. (2018) paper (also using mm/year but somewhat different scale). Irrigation can also be checked for reference with the same figure, but values may differ locally because irrigation is prognostically calculated by the model in CLM. Finally, we do translate the area values [mm/year] to volumetric [km3/year] (e.g. in figure 6 of our paper, and we confirm that our values are comparable to previous studies).

At the same time, we agree with the reviewer assessment that in many cases the irrigation or combined sectoral requirements in certain regions are significantly larger than the available streamflow. But this is not unusual. In such cases, the supply of the water comes from alternative sources, such as groundwater, reservoirs, large scale supply networks (e.g. larger than the gridcell in our simulations), inter-basin transfers, desalination or recycled water. Of course, there are also situations when there is simply not enough water, and then a shortage emerges with the subsequent consequences for the society.

> **Reviewer 2 Comment 4**
>
> Line 100-104: Is the main channel grid cell-scale?

**Response** Yes, each MOSART gridcell has one main channel. According to the CLM5/MOSART Documentation:

> MOSART divides each spatial unit such as a lat/lon grid or watershed into three categories of hydrologic units: hillslopes that convert both surface and subsurface runoff into tributaries, tributaries that dis- charge into **a single main channel**, and the main channel that connects the local spatial unit with upstream/downstream units through the river network.

At the same time, since the MOSART grid is a 0.5x0.5 while the CLM5 is 0.9x1.25 in our case, there will be several main channels which could be used for water supply at CLM5 grid cell level. Because of this, the remapping of sectoral fluxes from one model to the other is necessary.

To better reflect these points, we updated the text:

> River water availability within CLM5 is provided by the MOSART routing model. It utilizes a kinematic wave approach, providing information on varying channel velocities, water depth in channels, and channel surface water variations (Li et al., 2013). In its functionality, surface runoff from CLM5 first traverses hillslopes before merging with subsurface runoff and moving to a tributary network, finally ending up in the main channel (Fig. 1 from Li et al. (2013)). Each MOSART grid cell has a single main channel that connects the local spatial unit with upstream/downstream units through the river network. It is this main channel's water storage, aggregated at CLM5 grid-cell level, that are used to estimate current river water availability. It should be noted that the CLM5 and MOSART models can run using different grids, which is the case in this study, with MOSART running on a 0.5x0.5 grid, and CLM5 on 0.9x1.25 grid. This means that for a given CLM5 gridcell, several MOSART main channels will be sourced for water supply. The handling of these spatial discrepancies is done through remapping procedures in the coupler.

**Reviewer 2 Comment 5**

Line 125 mentions that the spatial scales of CLM5 and MOSART are different. From what I read from the paper, the MOSART seems to be a macro grid-scale river routing module relying on the grids defined in running CLM5. So I think there shouldn't be a scale mismatch between them?

**Response** We updated the text:

> During the coupling process, each CLM5 grid cell sends through the coupler to MOSART the information about how much water should be withdrawn and how much should be recycled back for each sector. The difference between the withdrawal and recycled part is the sectoral consumption, which is the net water amount which is transported from the river system to the land component. The CLM5 and MOSART spatial organization is different in this study, with CLM5 running on a 0.9x1.25 grid, while MOSART on a 0.5x0.5 grid. This needs to be taken into account when passing sectoral fluxes or water storage information from one model to the other during the coupling process.

**Reviewer 2 Comment 6**

Line 127-130: These sentences need a clarification. Does land unit mean the same with spatial unit? It is mentioned that the sub-grid variability is handled by snow/soil column and PFTs. Are these the same spatial concepts with lat/lon grid cell or watershed?

**Response** We agree and updated the text:

> In CLM, spatial land surface heterogeneity is represented through a nested subgrid hierarchy (Fig. A1) (Lawrence et al., 2019). Each grid cell contains multiple land units, columns, plant functional types (PFTs), and crop functional types

(CFTS, if crop option is on). Land units, capturing the broadest patterns, include glacier, lake, urban, vegetated, and crop. Urban units are further divided into density classes. Columns represent variability within a land unit, such as different soil and snow states. Vegetated units may have multiple columns for soil profiles, while managed vegetation units have irrigated and non-irrigated columns. Columns have up to 25 layers for ground and 10 for snow, which allows solving for water storage and snow dynamics. The PFTs and CFTs corresponding to the third subgrid level, referred to as patches, represent various trees, shrubs, grass and crops covers that populate the given region (Lawrence et al., 2019). The patch level is intended to capture the biogeophysical and biogeochemical differences between broad categories of plants in terms of their functional characteristics. While the subgrid heterogeneity is captured by the model in the sense of realistic fractions of different land units, PFTs and CFTs, their exact relative location is not represented. The calculations are done individually over each column and the outputs are then aggregated at grid cell level before exchanging information with the coupler.
* * *
**Reviewer 2 Comment 7**

Line 139: What is VOLR here?
* * *
**Response** VOLR is the variable name in the CLM5 model for the total water storage of the river network corresponding to the given gridcell. We updated the text to better reflect this point:

> For example, if MOSART has two active main channels within a CLM5 grid cell with a total water storage VOLR (corresponding variable name in the model), with the larger channel containing 80% of VOLR, and the smaller channel the remaining 20%, then the sectoral fluxes from the CLM5 grid cell will be distributed across the two available channels in the same proportion (i.e., 80%/20%).
* * *
**Reviewer 2 Comment 8**

Line 138: Why is the total water storage decided only by the channel water? Can you verify how the total water storage of a grid in CLM5 is estimated? It is need to check if soil water and groundwater are considered in the total water storage.
* * *
**Response** We do mention in Sect 2.1 that we use the model setup when only river water is used to supply the sectors:

> The default source for water supply for irrigation is the river network, with a user-defined possibility to supply from groundwater. At the moment, however, the simulated groundwater abstractions for irrigation are not constrained by observations and this new CLM5 module has not been thoroughly tested. Therefore, in this study, we exclusively use the default configuration where water is abstracted from the river network.

Concerning the soil water, it will have an influence on river water availability as CLM and MOSART are coupled. By resolving soil moisture dynamics, CLM calculates the subsurface

and surface runoff which are provided as inputs to MOSART that convert it to streamflow (CLM5 Documentation).

In principle, it would be possible to allow groundwater abstractions for the irrigation sector, but this feature was not yet tested, and such support was not yet added for the new sectors. But we agree that it will be more correct if other sources are implemented, which we discuss in more details in the Sect. 4 on Limitations and a way forward.

> **Reviewer 2 Comment 9**
>
> Line 144: Are there any reasons for this kind of order in allocating water?

**Response** We adopted the same order as currently done in H08 and VIC5 models, but also used in some other studies. The main argumentation for this is usually that this is a pragmatic choice taking into account the current lack of data on prioritization. In addition to this, the current order of water withdrawal reflects general premise that priority should be given to basic human needs and high value-added products in resources allocation. Municipal, industrial, and agricultural water use intensities per value added (service, manufacturing and power generation, and agricultural sectors) are estimated at 0.012, 0.063, and $2.2 \times 10^6 \ m^3$ per $10^6$ USD, respectively (Hanasaki et al., 2018). At the same time, in Rathore et al., 2024, in addition to this default prioritization they explored an alternative where agricultural demands go first. This resulted in a significant increase in unmet demands for municipal (domestic) and industrial sectors. If we would repeat the experiment, we would find the same results as can be inferred from the Appendix figures D2-D6. To better reflect these nuances we updated the text:

> Under this system, when water is scarce, it is allocated to sectors in the following priority order: domestic, livestock, thermoelectric, manufacturing, mining, and irrigation. Similar sectoral priority orders have been implemented in some Global Hydrological Models (GHMs; e.g., H08, Hanasaki et al. (2018) and VIC-5, Droppers et al. (2020)). This order reflects a general premise that priority should be given to high value-added products in resource allocation. Municipal, industrial, and agricultural water use intensities per value added are estimated at 0.012, 0.063, and $2.2 \times 10^6 \ m^3$ per $10^6$ USD, respectively (Hanasaki et al., 2018). This highlights that sectors such as municipal and industrial services provide higher economic returns per unit of water used compared to agriculture.

We also update the text in the Sect. 4 on Limitations:

> In our model, sectoral water use priorities are currently fixed and follow the order (in decreasing priority) domestic, livestock, thermoelectric, manufacturing, mining, and irrigation. While a similar hierarchy was also implemented in some other models (Hanasaki et al., 2018; Droppers et al., 2020), in reality the priority may vary based on regional circumstances, weather, policies, or changing socio-economic conditions. For example, a recent study suggests that in many regions the domestic and irrigation sectors often receive higher priority than other sectors during periods of droughts, heat waves, and compound hot-dry extremes (Cárdenas Belleza et al., 2023). At the same time, regional exceptions are possible, highlighting the need for more flexible approaches in modelling sectoral competition in GHMs and LSMs models (Cárdenas Belleza et al., 2023). For example, a recent study explored an alternative prioritization where agricultural demands are placed first (Rathore et al., 2024). This scenario resulted in around 30% increase in unmet demands for municipal (domestic) and industrial sectors for urban areas. If we were to replicate this experiment, similar results would be anticipated, as evidenced by the figures in Appendix D2-D6 where many gridcells experiencing water scarcity for irrigation sector, do not experience such scarcity for the other sectors. This further supports the development of more flexible prioritization schemes to study related uncertainty in unmet sectoral demands.
* * *
**Reviewer 2 Comment 10**

Line 169: This sentence needs more clarifications.
* * *
**Response**

We added some more clarifications, and specifically the contrast between non-irrigative (applied throughout the whole day) and irrigation (applied over 4 hours only):

> It should be mentioned here that the total consumed flux is not applied on surface soil all at once, but dribbled out evenly during the modelled day. This is in contrast to irrigation, where the total withdrawal is distributed uniformly over a period of 4 hours starting with 6AM local time.
* * *
**Reviewer 2 Comment 11**

Line 205: A sudden change in 'expected' or 'actual' withdrawal?
* * *
**Response** Thanks for noticing, we updated the text:

> Certain sectors may exhibit a sudden increase or decrease in the expected withdrawal amount at the onset of a new month.
* * *
**Reviewer 2 Comment 12**

Line 229: Is 2 year spin-up enough? Can you show how the steady-state of the model, especially concerning about the new sectoral water module, is confirmed (e.g., criteria)? What (water-related) variables were considered to confirm the model's steady-state?
* * *
**Response** The spin-up length for the CLM5 will depend on what exactly we are interested in. For example, to reach a quasi steady state for the carbon cycle, we might need a few hundred years for the spinup (Lawrence et al., 2019). In our case, the new sectoral water module results are sensitive mostly to the routing component state. From our experience in this case, 2-years are enough (Vanderkelen et al., 2022). At the same time, we did confirm this during the analysis. For example, when we looked at the global Actual vs Expected withdrawal ratio, we found a visible anomaly with higher unmet demands for the first year or so (see black box in Illustration 9). Locally, this translated for example in having unmet demands in regions where this shouldn't be the case. After excluding the first 2 years as

spinup, the anomalies are resolved and the model results seem correct both for global and grid cell metrics.

[Figure]

Illustration 9:

**Reviewer 2 Comment 13**

Figure 3: How the withdrawal is differ from the consumption? Is it explained before? Are these expected withdrawal or actual? All these information must be clarified. Also, how does this figure explain the sentence (line 245) saying that remapping procedure is found to be conservative?

**Response** We now added a definition for these terms at the beginning of the Sect. 2.2:

> The primary focus of our module development is to accurately depict the withdrawal and consumption of water across a variety of sectors. We define withdrawal as the gross amount of water removed from a water source for use in a particular sector. Sectoral water consumption, on the other hand, is the portion of water withdrawn that is actually consumed and not returned to the water source. It includes water that is lost through evapotranspiration, incorporated into products or crops, or otherwise not returned to the immediate water environment. This is achieved using a data-driven approach.

Concerning Fig. 3 which shows a close match between the original and remapped data (now in the Appendix) we write:

> In general, the remapping procedure is found to be conservative (Fig. 3), with relative errors of about 1-2% between the original and remapped data explained by upscaling effects when passing to a lower resolution land mask (Fig. B1).

**Reviewer 2 Comment 14**

Figure 4: It is very hard to tell the differences among the plots. Moreover, there is no color bar so the color gradients in the plots do not make any sense.

**Response** The objective of Fig. 4 (now in Appendix) was to only show that no withdrawal happens for the sector lower in priority, when the one higher in priority is not fully satisfied. Also, we don't have a colour gradient in this figure, but we now realized that it looks like this. So we added the following information to the figure caption to clarify this point:

> Evaluation of the sectoral competition algorithm, with each point representing a daily value at grid cell level. The points are plotted semi-transparently (alpha=0.5), therefore the more intense coloured parts simply indicates a larger concentration of values in that range. The plot was made by sampling the first 30 days of the year 2000 from the SectorWater experiment. The intersection between the unsatisfied sectoral withdrawal of the sector higher in priority and the actual withdrawal of the sector lower in priority represents the 0 value.

**Reviewer 2 Comment 15**

Line 338-340: This sentence seems to be unnecessary.

**Response** We opted to take the sentence out.

**Reviewer 2 Comment 16**

While I understand the concept of two-way interactions between the land and river in this model, this is just a simple water balance and, moreover, the watersheds' hydraulic gradient-driven nature is not explicitly considered. For example, the withdrawal of water from river should have an effect on surface/subsurface runoff. While I am saying this to change the structure of the model, but the authors should indicate the limitation of their two-way method applied in this work.

**Response** We appreciate the opportunity to clarify the capabilities of our model. We have considered each of the reviewer points carefully and would like to provide the following clarifications:

1) The CLM5 model handles multiple hydrological processes including evapotranspiration, infiltration, soil moisture dynamics, and plant water uptake. These processes are dynamically linked to the river routing performed by MOSART, which simulates river flow based on land surface inputs and changing river storages. This coupling allows for an advanced representation of water balance that incorporates feedback mechanisms between terrestrial and aquatic systems, rather than a simple input-output water balance model.

2) On the hydraulic gradient-driven nature of watersheds, while we indeed did not mention this in the text, significant progress was made in this regard for the CLM5 model. In particular we would like to refer to the Swenson et al. (2018) study. The research focused on improving the representation of subsurface water flow within hillslopes, which is a critical process affecting soil moisture, groundwater recharge, and streamflow. Traditionally, Earth system models, including the Community Land Model (CLM), have used simplified approaches to simulate hydrologic processes. These approaches often fail to capture the complexity of lateral subsurface flows that occur within hillslopes. Such flows are significant because they contribute to the redistribution of water and energy within the landscape, influencing various ecological and hydrological outcomes. In their study, Swenson et al. introduced a new

parameterization for intra-hillslope lateral subsurface flow in the CLM. This involved integrating detailed processes that account for the movement of water through both shallow and deeper soil layers across different landscape positions, such as ridges and valleys. The enhanced model resulted in better simulation of the flow of water from higher elevations to lower areas, improving predictions of soil moisture dynamics and streamflow.

3) Concerning the impact of water withdrawal on river flow and subsequently on surface and subsurface runoff, this is actually captured by our model. For example, through the withdrawal of irrigation or consumption of non-irrigative sectors, the river flow is reduced in most regions (Illustration 7). Also, by applying the consumption or irrigation water on the surface soil, the water becomes part of the surface water balance, therefore contributing to infiltration, evapotranspiration, surface and sub-surface runoff.

We hope that these details help clarify the model capabilities. But we also agree with the reviewer, that further improvements are needed and for this reason we tried to be as exhaustive as possible on the future next development steps in the limitation section.

[revised manuscript text omitted]

Swenson, S. C., Clark, M., Fan, Y., Lawrence, D. M., Perket, J. (2019). Representing intra-hillslope lateral subsurface flow in the community land model. Journal of Advances in Modeling Earth Systems, https://doi.org/10.1029/

---

## Referee Report (RR1)

Title: Bridging the gap: a new module for human water use in the Community Earth System Model version 2.2.1

**Summary**
The paper presents a new module for human water use in CESM. This module integrates sectoral water abstractions for multiple sectors, conserving water by integrating abstractions from the land component with river component flows and dynamically calculating daily water scarcity based on local demand and supply. The findings emphasize the importance of including all sectors for water scarcity assessment capabilities and highlight areas for potential future refinement.

Overall, the paper makes a significant contribution to the field of Earth system modeling by enhancing the representation of human water use in CESM. The detailed methodology, comprehensive validation, and insightful analysis make it a valuable resource for researchers and policymakers interested in sustainable water management. However, I found adding some more discussion could improve the manuscript.

I was invited to review this paper in the second round. Please let me know if there are any conflicts with comments or suggestions raised in the first round.

**Major comments:**
1. Validation: The paper is well-written and novel in that the module integrates multiple sectors, providing a holistic view of water use and scarcity. The model is validated against historical data and known water scarcity hotspots. However, more validation could be done, for example, using stream gauge data, evapotranspiration data, and satellite land surface temperature datasets. Alternatively, it could be discussed whether such validation will be done in future work and how it would benefit the model.

2. Groundwater Abstractions: The model currently focuses only on river water abstractions, potentially underestimating groundwater use in arid regions. The authors might need to discuss future model development plans or explain why river water abstractions are more important than groundwater use. The study found that non-irrigative sectoral consumption has an insignificant effect on regional climate. Could this be because the study neglects groundwater use?

3. CESM Coupling: In this paper, only offline CLM simulations have been done, but the title mentions "CESM." The authors might need to discuss whether there are future plans to use this new module in coupled CESM simulations and what potential issues might arise when coupling with atmospheric or other models.

4. Introduction Enhancement: The introduction could benefit from an overview of global hydrological models (e.g., WaterGAP, GHM, PCR-GLOBWB), including whether and how human water use has been modeled and what the limitations are compared to land

surface models (LSMs). This is particularly important since there are discussions on GHMs in the Results and Discussion sections but not in the Introduction.

**Minor comments:**
[The line number refers to the version without tracked change.]

Line8: have-> has
Line 61 "the" land-use and land-cover change (LULCC)
Line 77: Maybe change "is focused" to "focuses"
Line 105: It might be worth mentioning why CLM on a 0.9x1.25° grid is chosen. Is it for future application in coupled CESM, or to match the input data? Also, please add "°" throughout the manuscript.
Line 111: have-> has
Line 113: Is "missing part" the "shortfall"?
Line 140: CFTs
Line 163: "The" same approach
Line 164: being-> is
Line 183: indicating-> meaning?
Line 185: will depend-> depends
Line 186: little losses-> few losses
Line 190: is -> are
Line 192: What if the land grid consists only cropland and/or urban?
Line 299: show?

- How does the new module deal with iced rivers/iced soil if there is human water use?
- Be consistent with "grid cell" or "gridcell." Are they referring to different things?

---

## Author Response (AR2)

**Bridging the gap: a new module for human water use in the Community Earth System Model version 2.2.1**

Geoscientific Model Development

August 24, 2024

S. I. Taranu[1], D. M. Lawrence[2], Y. Wada[3,4], T. Tang[3,4] E. Kluzek[2], S. Rabin [2] Y. Yao[1], S. J. De Hertog[1,5], I. Vanderkelen[6,7,8], and W. Thiery[1]
sabin.taranu@vub.be

[1] Vrije Universiteit Brussel, Department of Water and Climate, Brussels, Belgium
[2] National Center for Atmospheric Research, Climate and Global Dynamics Laboratory, Boulder, CO, USA
[3] Biological and Environmental Science and Engineering Division, King Abdullah University of Science and Technology, Thuwal, Saudi Arabia
[4] International Institute for Applied Systems Analysis, Laxenburg, Austria
[5] Universiteit Gent, Department of Environment, Q-ForestLab, Ghent, Belgium
[6] Wyss Academy for Nature at the University of Bern, Bern, Switzerland
[7] Climate and Environmental Physics division, University of Bern, Bern, Switzerland
[8] Oeschger Centre for Climate Change Research, University of Bern, Bern, Switzerland

**Contents**

**1   Reviewer 1**

We thank the Reviewer 1 for the constructive feedback to improve the manuscript. Below, we address every comment carefully and explain the corresponding changes in the manuscript.

**1.1   Major comments**

> **Reviewer 1 Comment 1**
>
> Validation: The paper is well-written and novel in that the module integrates multiple sectors, providing a holistic view of water use and scarcity. The model is validated against historical data and known water scarcity hotspots. However, more validation could be done, for example, using stream gauge data, evapotranspiration data, and satellite land surface temperature datasets. Alternatively, it could be discussed whether such validation will be done in future work and how it would benefit the model.

**Response** We agree that in the future, the CESM/CLM community would benefit from such analysis. In this case, we considered adding such evaluation, especially using stream gauge data to see if adding sectoral abstractions would improve streamflow in highly managed rivers. But in the end, we decided to delay this till the next version of the model is released because of some documented issues regarding relevant fluxes (e.g. runoff), and since model recalibration in the case of LSMs/ESMs is very challenging. We now explain our reasoning more in the text:

> While more model development may be needed to represent relevant processes related to human-water interactions, another important aspect to consider is model evaluation and calibration for hydrological variables. The variables which are the most important for water availability modelling are precipitation, evapotranspiration, snowpack dynamics, glacial melt, soil moisture, surface runoff, river flow, and groundwater levels and recharge. For example, Vanderkelen et al. (2022) showed that while globally the runoff biases in CLM5 are very small (+0.077 mm/day), large regional biases exist. Aggregated at the level of a catchment, such biases can result in significant river discharge biases, limiting the model usability for water management purposes (Mizukami et al., 2021). Efforts are being made to solve this problem with targeted evaluation studies to understand hydrological parameter uncertainty in CLM5 (Yan et al., 2023). At the same time, more efficient and transparent objective calibration protocols to improve model performance for a given set of targets are being developed (Dagon et al., 2020; Cheng et al., 2023). Unfortunately, running large parameter perturbation ensembles for sensitivity testing and application of objective calibration protocols remains very expensive for LSMs/ESMs, and are usually done only for the release versions of the model. In the future when the model is calibrated, it could be interesting to expand our analysis by assessing the added value of the implementation of human water management on river flow and other relevant hydrological variables. For example, in the case of GHMs, it was found

that considering human related impacts, including land-use-change, reservoir operations and water abstractions results in a general performance increase to represent streamflow and hydrological extremes (Veldkamp et al., 2018).

> **Reviewer 1 Comment 2**
>
> Groundwater Abstractions: The model currently focuses only on river water abstractions, potentially underestimating groundwater use in arid regions. The authors might need to discuss future model development plans or explain why river water abstractions are more important than groundwater use. The study found that non-irrigative sectoral consumption has an insignificant effect on regional climate. Could this be because the study neglects groundwater use?

**Response** This is an interesting suggestion which is worth investigating. We now added to the text:

> Our findings show that only irrigation has the potential to significantly affect local climates (for scales above a 100 km), while the effect of non-irrigative sectors is negligible. This might not be true at higher resolutions, especially if we would consider groundwater abstractions and land-atmosphere coupling. For example, Keune et al. (2018) study reveals that groundwater abstraction can significantly weaken the continental sink for atmospheric moisture by reducing soil moisture and altering surface energy fluxes. This reduction in soil moisture leads to decreased evapotranspiration, which in turn can diminish the local recycling of moisture back into the atmosphere. The diminished recycling can lead to reduced precipitation in some regions, thereby exacerbating local drought conditions. Furthermore, the weakening of the continental moisture sink due to groundwater depletion can have far-reaching implications for weather patterns and regional climate stability. While we find that the climatic impacts of other sectors like domestic and industrial water use are comparatively small, their inclusion in the model remains important for water scarcity assessment capabilities.

When it comes to groundwater use in our model, we do recognize that this is one of the main limitations, and we discuss this on multiple occasions in the text, inviting further model development and studies on the subject:

[revised manuscript text omitted]

> **Reviewer 1 Comment 4**
>
> Introduction Enhancement: The introduction could benefit from an overview of global hydrological models (e.g., WaterGAP, GHM, PCR-GLOBWB), including whether and how human water use has been modeled and what the limitations are compared to land surface models (LSMs). This is particularly important since there are discussions on GHMs in the Results and Discussion sections but not in the Introduction.

**Response** We now extended the introduction, by mentioning the GHMs capabilities when it comes to human water management. While we cover the relevant literature for all main models, we don't really go into details on individual models because there are many differences between each of them, and we think that this is well covered already in Telteu et al. 2021 (`https://doi.org/10.5194/gmd-14-3843-2021`) and more recently in Müller Schmied et al. 2024 (`https://doi.org/10.5194/egusphere-2024-1303`):

To further contribute to the effort of improving human-water interactions in LSMs/ESMs, we here present a new sectoral water use module for the Community Earth System Model version 2 (CESM2). Our data-driven module advances the representation of human water use by incorporating a comprehensive account of water abstractions for domestic, livestock, thermoelectric, manufacturing, and mining sectors, thereby complementing the existing irrigation module (Lawrence et al., 2019). Through this development, the CESM2 model and its land component approaches more the capabilities of state of the art Global Hydrological Models, that not only represent essential hydrological processes, but also commonly integrate human-related water management practices, including reservoir operations, water abstractions, pollution, and the exploration of alternative water sources like desalination and wastewater reuse (Hanasaki et al., 2016; Sutanudjaja et al., 2018; Hanasaki et al., 2018; Burek et al., 2020; Droppers et al., 2020; Müller Schmied et al., 2021; Van Vliet et al., 2021; Jones et al., 2023). Additionally, it enables fully coupled applications, allowing for the exploration of

feedbacks between human water use and land-atmosphere interactions (Keune et al., 2018), which is not achievable with GHMs.

**1.2 Minor comments**

> **Reviewer 1 Comment 5**
>
> 1. Line8: have-> has
> 2. Line 61 "the" land-use and land-cover change (LULCC)
> 3. Line 77: Maybe change "is focused" to "focuses"
> 4. Line 111: have-> has
> 5. Line 113: Is "missing part" the "shortfall"?
> 6. Line 140: CFTs
> 7. Line 163: "The" same approach
> 8. Line 164: being-> is
> 9. Line 183: indicating-> meaning?
> 10. Line 185: will depend-> depends
> 11. Line 186: little losses-> few losses
> 12. Line 186: little losses-> few losses
> 13. Line 190: is -> are
> 14. Line 299: show?

**Response** We thank the reviewer for the careful read. We corrected these mistakes, and proofread the article for any other mistakes.

> **Reviewer 1 Comment 6**
>
> Line 105: It might be worth mentioning why CLM on a 0.9x1.25° grid is chosen. Is it for future application in coupled CESM, or to match the input data? Also, please add "°" throughout the manuscript.

**Response** We now added to the text:

> The simulations were run with a horizontal resolution of 0.9x1.25° and a default 30-minute time step. While it would have been possible to run simulations at higher resolutions (e.g. 0.5x0.5°), we opted for the 0.9x1.25° grid because it is one of the two scientifically supported grids for the IHISTCLM50BgcCrop configuration. In future applications, where the focus extends beyond demonstrating the module's capabilities, a higher resolution setup may be preferred to provide more detailed regional insights.

We also added now "°" where needed.

> **Reviewer 1 Comment 7**
>
> Line 192: What if the land grid consists only cropland and/or urban?

This might probably happen only at very high resolutions. In general, the land cover is highly heterogenous. In our case, since the resolution is of the order of 100 km there will be

no grid cell which would have a single land unit.

At the same time, if this would happen and there will be no natural vegetation, this might cause some issues because it wouldn't be possible to correctly process the consumption flux. At the moment, the code for this development is pending for being integrated into the source. We added your comment to the pull request discussion on GitHub, so in case this the development will be integrated, this will be taken care of (likely through the addition of some exception case, e.g., if no natural vegetation, do this). We thank the reviewer for highlighting this potential issue.

> **Reviewer 1 Comment 8**
>
> How does the new module deal with iced rivers/iced soil if there is human water use?

We now added to the text:

> The information about how much water is potentially available for sectoral use is provided by the coupler to the CLM5 model at grid-cell level by calculating the corresponding total river network storage in the MOSART model. This includes only the liquid water from the rivers, excluding iced river water or the water stored directly in the soils, which are not used to meet sectoral demands.

In the routing model, only the liquid water runoff contributes to VOLR. For the ice routing, a different variable is present. Therefore, if in a given grid cell all the river water is frozen, VOLR will be zero, and no abstractions will occur. About iced soils, this also doesn't cause any issues, because we do not take the water from the soil, but only from the liquid water routed in the rivers (VOLR).

Therefore, the module is robust and will continue working as expected for both iced rivers/soils.

> **Reviewer 1 Comment 9**
>
> Be consistent with "grid cell" or "gridcell." Are they referring to different things?

**Response** Thanks for noticing, we now keep the term consistent — "grid cell".

[revised manuscript text omitted]

Swenson, S. C., Clark, M., Fan, Y., Lawrence, D. M., Perket, J. (2019). Representing intra-hillslope lateral subsurface flow in the community land model. Journal of Advances in Modeling Earth Systems, https://doi.org/10.1029/

Veldkamp, T.I.E., Zhao, F., Ward, P.J., De Moel, H., Aerts, J.C., Schmied, H.M., Portmann, F.T., Masaki, Y., Pokhrel, Y., Liu, X. and Satoh, Y., 2018. Human impact parameterizations in global hydrological models improve estimates of monthly discharges and hydrological extremes: a multi-model validation study. Environmental Research Letters, 13(5), p.055008.

Hanasaki, N., Yoshikawa, S., Kakinuma, K., and Kanae, S.: A seawater desalination scheme for global hydrological models, Hydrol. Earth Syst. Sci., 20, 4143–4157, https://doi.org/10.5194/hess-20-4143-2016, 2016.

Jones, E. R., Bierkens, M. F. P., Wanders, N., Sutanudjaja, E. H., van Beek, L. P. H., and van Vliet, M. T. H.: DynQual v1.0: a high-resolution global surface water quality model, Geosci. Model Dev., 16, 4481–4500, https://doi.org/10.5194/gmd-16-4481-2023, 2023.

Keune, J., Sulis, M., Kollet, S., Siebert, S., Wada, Y. (2018). Human water use impacts on the strength of the continental sink for atmospheric water. Geophysical Research Letters, 45, 4068–4076. https://doi.org/10.1029/2018GL077621